**Data Availability Statement:** The data underlying the results presented in the study are available

# Explaining socioeconomic inequalities in self-reported health outcomes: The mediating role of perceived life stress, financial self-reliance, psychological capital, and time perspective orientations

**Karen Schelleman-Offermans** *, Karlijn Massar

Department of Work and Social Psychology, Maastricht University, Maastricht, Limburg, The Netherlands

* Karen.Offermans@maastrichtuniversity.nl

## Abstract

### Objective

The main aim of the current study was to investigate what role perceived life stress, psychological capital (PsyCap), financial self-reliance and time perspective orientations play in explaining socioeconomic health inequalities, specifically self-perceived health and self-reported physical health conditions.

### Methods

Individuals (total n = 600) aged 16+ years from a general Dutch population sample (LISS panel) completed an online questionnaire measuring three different SEP indicators (highest achieved educational level, personal monthly disposable income and being in paid employment), perceived life stress, PsyCap, financial self-reliance, time perspective, self-perceived health, and self-reported physical health conditions. Structural equation modelling using a cross-sectional design was used to test the mediation paths from SEP indicators to self-perceived health and self-reported physical health conditions through perceived life stress, PsyCap, financial self-reliance and time perspective orientations.

### Results

Highest achieved educational level and being in paid employment showed to play a role in the social stratification within self-reported and self-perceived health outcomes, whereas this was not found for personal monthly disposable income. The association between a lower highest achieved educational level and lower self-perceived health was mediated by lower PsyCap and higher perceived life stress levels. The association between a lower highest achieved educational level and higher levels of self-reported physical health conditions was mediated by less financial self-reliance and higher perceived life stress levels. Although no mediating role was found for time perspective orientations in the association

from CentERdata, Institute for data collection and research (https://www.lissdata.nl/; contact via info@lissdata.nl; data accession name "DATAfile PLOSONE_Schelleman-Offermans.anonymous"). The syntax files (SPSS and Mplus) regarding the analyses of this study for interested researchers who want to replicate the findings are available from (https://osf.io/qntjd/). The authors confirm they had no special access privileges to the data.

**Funding:** This study was funded by the Netherlands Organisation for Health Research and Development (https://www.zonmw.nl/en/), grant number 531001410, awarded to KM. The LISS panel data were collected by CentERdata (Tilburg University, The Netherlands) through its MESS (Measurement and Experimentation in the Social Sciences) project funded by the Netherlands Organization for Scientific Research (https://www.nwo.nl/en). The funders had no role in the study design, data collection and analysis, decision to publish, or preparation of the manuscript.

**Competing interests:** The authors have declared that no competing interests exist.

between the measured SEP indicators and health outcomes, negative time perspective orientations were associated with either self-perceived health or self-reported physical health conditions.

## Conclusions

reserves (PsyCap and financial self-reliance) and perceived life stress seem to play a larger role in explaining the health gradient in achieved educational level than time perspective orientations. Prevention efforts trying to reduce the SEP-health gradient should focus on a) increasing reserves and lowering perceived life stress levels for individuals with a low achieved educational level, and b) reducing unemployment and narrowing opportunity gaps in education for people with a low SEP.

## Introduction

There are social, political and economic forces that shape the nature of social and structural relations in the society in which we live resulting in an unequal distribution of money, power and resources [e.g., 1–3]. Education, income and wealth are indicators of what resources individuals hold and what sort of 'life chances' they have [e.g., 1–3]. These structural positions are powerful determinants of the likelihood of health-damaging exposures and of possessing particular health enhancing resources. For instance, a higher income allows increased access to better quality material resources (e.g., food, housing, access to services) and a higher educational level provides individuals with greater knowledge and skills which can provide a higher social standing, self-esteem and facilitate participation in society [1]. The most commonly used indicators for socio-economic position (SEP) are therefore a) income level, b) educational level, and c) paid employment. Indeed, there are large differences in health behaviours between individuals within advantaged groups with a higher SEP and people within less advantaged groups with a lower SEP, leading to high disparities in self-rated health, health status as well as life expectancy amongst these groups [4–7]. To decrease the social stratification of health inequities, the obvious fundamental option is to change its structural drivers such as decreasing inequities in power, money and resources determined by the macro socioeconomic and political context [2]. Nevertheless, another option is to change specific risk or protective factors mediating the effect of SEP on health. Conventional explanations for the SEP-health gradient such as differences in stress, childhood circumstances and health behaviours are well established [8], yet only explain part of the SEP-Health gradient. Efforts to gain insight into the psychological mechanisms explaining health inequalities have lagged behind. Clarifying the role of psychological variables in explaining health disparities may provide important knowledge to design targeted intervention strategies that may reduce the effects of SEP on health outcomes and perceptions. The main aim of the current study is to gain more insight into the psychological mechanisms explaining the SEP-health gradient. The current study utilizes the reserve capacity model (RCM) [9] as a theoretical basis to gain insight into the psychological mechanisms of health inequalities in a general Dutch sample. More specifically, this study investigates the role of perceived life stress, perceived intra-personal (psychological capital) and tangible (financial self-reliance) reserves and time perspective orientations in explaining differences in self-reported physical health conditions and self-perceived health in individuals with a lower and higher SEP. Self-rated or perceived health covers a variety of health outcomes

and has shown to be consistently associated with morbidity and mortality [6, 7]. This suggests that a single measure of self-rated health is a strong predictor of overall health status.

## The Dutch context

In the Netherlands, as in many other European countries, health inequalities between people with a high and low SEP exist; people with a low SEP show for instance a higher prevalence of chronic diseases and self-assessed poor health [10]. Also, individuals with a low SEP live on average 6 years shorter and 15 years with a less well-experienced health, compared with individuals with a high SEP in the Netherlands [11]. When looking at the more fundamental drivers of health inequalities, although being one the richest countries in the world, the Netherlands faces an income inequality of .28% (Gini coefficient) and a poverty rate of 8.8% [12]. Nevertheless, for European standards, the Netherlands has a relatively flat income distribution; the differences in disposable income between people are small. Also, overall unemployment (3.3% in 2017) is low in the Netherlands [13]. Although high quality education and student loans are available to every person growing up in the Netherlands, children of lower educated parents are less likely to go to university [14]. This, for some part, seems to be due to the lower expectations teachers have of these children and the resulting lower support schools give them in their attempts to access tertiary education, even when they have satisfactory test scores [14]. Furthermore, the Netherlands has a universal healthcare system, managed by the Dutch government and supplemented by private insurers. Everyone living or working in the Netherlands must obtain basic level health insurance from a Dutch provider. People with a low income have the right to apply for health care contribution supplied by the Dutch government.

**Explaining health inequalities using the reserve capacity model.** That large disparities in self-rated health, health status as well as life expectancy exist between individuals within advantaged (high SEP) and less advantaged groups (low SEP) is well established in scientific literature [e.g., 4–7]. The reserve capacity model is a framework explaining how a lower SEP results in health disparities over time, and it specifically explicates the mediating role of 'reserve capacities', stress and positive/negative cognitions and emotions in explaining the SEP and health status gradient [9, 15–17]. Three types of 'reserve capacities', defined as resources which individuals can use in times of need, are proposed by the RCM: tangible reserves (e.g., money and transportation) intrapersonal reserves (e.g., self-esteem and psychological capital) and interpersonal reserves (e.g., social support). The RCM also provides an explanation why the amount of reserves individuals have to their disposal differs depending on one's SEP. Research indicates that individuals with a low SEP often live in neighbourhoods with high poverty, increased crime rates, and fewer green areas such as parks [e.g., 18]. Since these types of environments place high demands on individuals due to perceptions of danger, urgency, and unpredictability, it is assumed that individuals with a low SEP therefore more often need to use their reserves and, as a result, more quickly deplete their reserves. Such environments indeed seem to result in behaviours that are reactively driven by environmental cues [19]. The RCM also posits that individuals with a low SEP experience more daily hassles and major stressors in their lives compared with individuals with a higher SEP, and also have fewer reserves available to them to cope with these stressors [9]. The lower reserves and increased stress levels that individuals with a low SEP experience are in turn predicted to increase their experience of negative emotions and cognitions, which can negatively impact on health outcomes [9]. Inferring from the theoretical assumptions of the RCM (see Fig 1), the main aim of the current study was to investigate whether the negative association between indicators of a low SEP and subjective self-reported health outcomes can be explained by higher perceived life

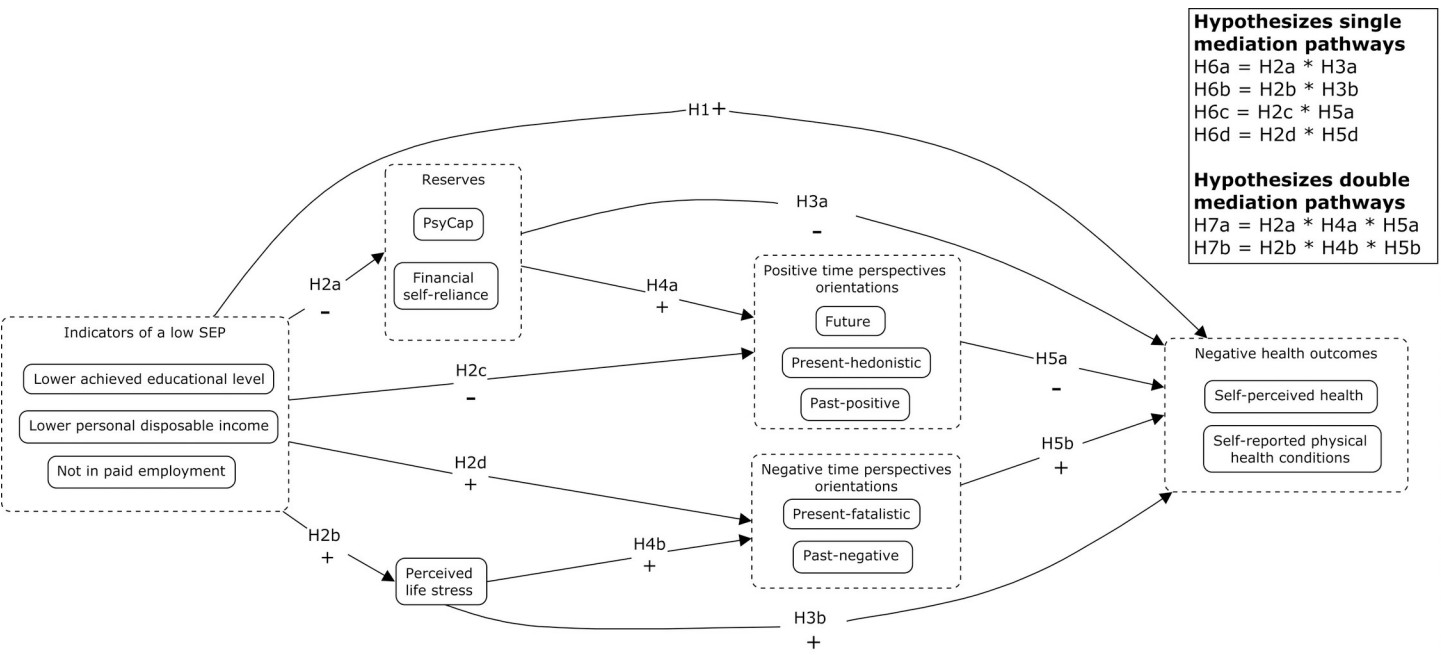

**Fig 1. Model and hypotheses tested in the current study.**

stress, lower intrapersonal and tangible reserves and consequently more negative or positive time perspectives (cognitions).

An intrapersonal reserve that can play an important in explaining health inequalities, which has been included in the current study, is Psychological Capital (PsyCap). PsyCap is a higher-order construct that consists of four psychological intrapersonal reserves–respectively hope, optimism, resilience and efficacy–that share a common core [20]. This common core is characterized by a focus on identifying one's strengths, making positive appraisals of one's chances of success, and perceiving that one's goals are within reach and under one's control. Briefly, Luthans, et al. [20] define PsyCap as an "[. . .] individual's positive psychological state of development, characterized by: (a) having confidence and skills to take on and put in the necessary effort to succeed at challenging tasks (efficacy); (b) making positive attributions about succeeding now and in the future (optimism); (c) persevering toward goals and, when necessary, redirecting paths to goals in order to succeed (hope); and (d) when beset by problems and adversity, sustaining and bouncing back and even beyond to attain success (resiliency)". Although the level of PsyCap can differ for different life domains (e.g., work or health), it has been established that having higher levels of these psychological resources or reserves is beneficial: Increased PsyCap levels positively influence job satisfaction and objective work outcomes, and increasingly, predict satisfaction with (social) relationships, and subjective and objective health outcomes [21]. For example, Luthans et al. [21] showed that increased health-related PsyCap is associated with lower BMI and cholesterol levels, as well as with health satisfaction. In turn, in a separate analysis, health satisfaction showed a significant association with overall well-being in this study. Also, PsyCap has been shown to be open for development. For instance, in a study by Rew, Powell, Brown, Becker & Slesnick [22], the feasibility and efficacy of a brief psychological capital intervention was examined using a quasi-experimental pre-post research design with repeated measures. The brief intervention aimed to reduce health-risk behaviors (alcohol use and sexual risk behavior) in 80 ethnically diverse homeless women by

increasing their psychological capital. Study results showed that within this group, substance use decreased significantly over time whereas safe-sex self-efficacy and behaviors significantly increased over time [22].

How financially self-reliant a person is, is a tangible reserve that may be important in explaining differences in subjective health perceptions between individuals scoring lower and higher on different SEP indicators. A higher financial self-reliance (e.g., being able to pay your bills independently) may indicate higher tangible reserves and in turn, in accordance with the RCM, may increase positive cognitions and emotions and increase health outcomes.

One cognitive factor that may be influenced by levels of stress and/or reserves and that has shown to motivate an individual's goal and to influence health outcomes are the relative temporal orientations that an individual holds, also referred to as time perspective orientations [23]. One's time perspective orientations describe how one's subconscious perception or weighing of the past, present, and future influences decision-making, including health-related decision making [24], and it has consequences for our physical and mental health [e.g., 25]. According to Zimbardo & Boyd [24] individuals can think about time in different ways, and they identified five distinct time perspectives: past-negative, past-positive, present-hedonistic, present-fatalistic, and future.

Present-oriented individuals tend to focus on the immediate (positive) consequences of their behaviour, whereas future-oriented individuals give more importance to the future consequences of such behaviours, even if there are immediate costs [e.g., 26–28]. Individuals with a predominantly past-oriented time perspective tend to relive past events, either positive or negative, and base decision making and behaviours on their appraisals of these events. Both a present-fatalistic time perspective (i.e., "a fatalistic, helpless, and hopeless attitude toward the future and life"; [24, p.1275]) and a past-negative time perspective (i.e., a negative attitude towards the past; [24]) have shown to be associated with several negative emotions, such as feelings of hopelessness and a lack of control over life. A future time perspective, on the other hand, causes the individual to regulate their behavior, establish goals and expectations, and to motivate and monitor performance [24].

Research indicates that time perspective orientations can consistently be linked to different health outcomes and life satisfaction [e.g., 23]. For instance, a present-fatalistic time perspective has been associated with lower levels of well-being and life satisfaction [25]. The higher probability of individuals with a low SEP to have low financials means and to live in areas with high demands [e.g., 18], may negatively influence their ability to make future plans. Previous research has indeed shown indications that individuals' time perspective orientations are socioeconomically patterned [29, 30]. Specifically, the high levels of environmental demands, the low levels of control over their environment (e.g., in their jobs), and low levels and rapid depletion of means or reserves that individuals with a low SEP experience, may foster a present-fatalistic and past-negative time perspective for low SEP individuals [19]. There are indications that individuals with a low SEP indeed hold beliefs that health outcomes are the result of predetermination and therefore are inevitable, and they think less about the future than individuals with a higher SEP [30, 31]. It then logically follows that if individuals believe that their future health is out of their personal control, they are less likely to perceive current health-protective behaviors as relevant or necessary, and will be less likely to engage in such behaviors. Thus, to increase positive health outcomes among groups with a lower SEP, it might be important to stimulate perceptions of a more positive future, by means of focusing on optimism and hope.

**Existing evidence for the proposed mediating pathways by the RCM.** In addition to the firmly established relationship between higher levels of stress in individuals with a low SEP (compared with higher SEP) and its negative effects on health [e.g., 32, 33], support for other

mediating pathways proposed by the RCM has been found by several studies for a variety of health-related outcomes. For instance, research by Gallo, Bogart, Vranceanu & Matthews [34] found that women with a low SEP compared with a high SEP showed lower levels of positive affect, an effect that was mediated by their lower levels of perceived control, and higher levels of social strain. Further, Bosma, Schrijvers, and Mackenbach [35] conducted longitudinal research among a Dutch general population sample and showed that the association between a low SEP and a higher mortality rate was mediated by deficient levels of control beliefs in individuals with a low SEP. Another longitudinal study [29] investigated the effect of childhood social class on adult self-perceived health and revealed that reserve capacities, operationalized as having an external locus of control and an absence of active problem-focused coping styles, explained half of the variance of this relationship. Furthermore, there is evidence for the mediating effect of lower reserve capacities–specifically, lower optimism, self-esteem or social support–and negative emotions on the relationship between SEP and metabolic syndrome [36]. Lastly, Bennett, Buchanan, Jones, & Spertus [37] showed that psychological reserves (cognitive-emotional factors) partially mediated the relationship between SEP and mental health status appraisals in patients suffering from myocardial infarctions. Moreover, the literature on the RCM indicates that psycho-cognitive variables (reserve capacities), stress and positive/negative cognitions and emotions can play a mediating role in explaining the (indirect) effect of SEP on various health outcomes, and as such illustrates the usefulness of the RCM in understanding the psychosocial pathways leading to socioeconomic health inequalities.

## The current study

As reviewed above, previous research [e.g., 17, 35] provided some evidence for parts of the RCM model. Nevertheless, no study up to now, to our knowledge, included stress, reserve capacities, positive and negative cognitions, and self-reported health outcomes in one model. The main aim of the current study is to investigate the psychological mechanisms underlying the SEP-health gradient, with a specific focus on perceived life stress, PsyCap (intra-personal reserve), the degree of financial self-reliance (tangible reserve) and time perspective orientations (positive and negative cognitions). The current study will therefore provide a more comprehensive insight into the psychological mechanisms underlying the SEP-Health inequity as well as the interrelationships between these variables.

To clarify the psychological mechanism underlying the SEP-health gradient as proposed by the RCM, several hypotheses were tested in the current study (see Fig 1 for a visual overview). First, we hypothesize that indicators of a lower SEP (i.e., a lower achieved educational level, lower personal disposable income and not being in paid employment) are directly associated with more negative self-reported health outcomes (higher self-reports of physical health conditions and lower perceptions of general health; H1). Also, indicators of a lower SEP are expected to be associated with lower reserves, i.e., self-reports of PsyCap and financial self-reliance (H2a), higher levels of perceived life stress (H2b), lower levels of positive times perspective orientations (past-positive, present-hedonistic and future, H2c) and higher levels of negative time perspective orientations (past-negative, present-fatalistic; H2d). Furthermore, a negative association is hypothesized between reserves (PsyCap and financial self-reliance) and more negative self-reported health outcomes (lower self-perceived health and more self- reported physical health conditions; H3a). A positive association is expected between reserves (PsyCap and financial self-reliance) and more positive time perspective orientations (H4a), but a higher level of perceived life stress is expected to be associated with more negative self-reported health outcomes (H3b) and more negative time perspective orientations (H4b). Positive time perspective orientations are, in turn, expected to be associated with more positive self-reported

health outcomes (H5a), whereas negative time perspective orientations with more negative self-reported health outcomes (H5b).

Moreover, the current study hypothesized that the association between indicators of a lower SEP and more negative self-reported health outcomes is mediated by several mediation paths. First, single mediation paths are hypothesized through either lower levels of reserves (H6a), higher perceived stress levels (H6b), lower levels of positive time perspective orientations (past-positive, present-hedonism, future; H6c) and/or higher levels of negative time perspective orientations (present-fatalism, past-negative; H6d). Secondly, double mediation paths are expected through higher perceived stress levels and a more negative time perspective orientations (higher past-negative and/or present-fatalistic time perspective; H7a) and/or through lower reserves and less positive time perspective orientations (lower levels of future, past-positive time and/or present-hedonistic time perspective; H7b).

## Method

Cross-sectional data were collected through the LISS (Longitudinal Internet Studies for the Social sciences) panel administered by CentERdata (Tilburg University, The Netherlands, www.centerdata.nl/en). The LISS panel is a representative sample of Dutch individuals who participate in monthly Internet surveys. The panel is based on a true probability sample of households drawn from the population register. Households that could not otherwise participate are provided with a computer and Internet connection [38]. A longitudinal survey is fielded in the panel every year, covering a large variety of domains including work, education, income, housing, time use, political views, values and personality. Relevant ethical safeguards were met with regard to participant confidentiality and written consent (CentERdata; www.centerdata.nl/en). Additionally, the Ethical Review Committee of Psychology and Neuroscience of Maastricht University approved the study protocol (Reference number 188_10_02_2018_S16). Self-perceived health and self-reported physical health conditions were collected during the regular panel measurements in November-December 2018 (Health wave 11). Data regarding psychological capital, financial self-reliance, time perspective orientations and perceived stress were collected three months later (February-March 2019) using a random sample specifically drawn for the current study. Data were analysed anonymously by Maastricht University.

### Procedure

A random sample was drawn of 600 potential respondents already participating the LISS panel study, stratified by sex and with an oversampling of individuals with a low SEP to ensure a sufficiently large group scoring low on SEP indicators in the final sample. Individuals scoring lower on SEP indicators were oversampled in such a way that half of this random sample scored low on the SEP indicator highest achieved educational level and personal disposable monthly income. There was a 79.8% response rate (N = 479) for the questionnaire administered in February-March 2019. All participants completed the items of the questionnaire administered in February-March 2019 and 19 participants (4%) had missing values on the health questionnaire (measuring self-perceived health and self-reported physical health conditions), which was administered in November-December 2018. The two datasets were combined into one cross-sectional dataset with a total sample of 600 participants whenever missing are imputed.

### Participants

In the analytic sample, a total of 255 men (42.5%) were present in the sample. The mean age was 53.4 years (SD = 18.79; range 16–96) for the complete analytic sample. No significant

difference ($F$ = 6.64, p = .10) in mean educational level was found between men and women. There were significantly more men in paid employment than women ($\chi^2$ = 10.96, df = 1, $p$ = .001). Also, participants in paid employment significantly ($F$ = 63.2, $p$ < .001) differed in mean age compared with participants not in paid employment; 45.9 (SD = 11.8, mean std. error = .8) for participants in paid employment and 57.1 (SD = 20.4, mean std. error = 1.0) for participants not in paid employment. The mean educational level did not significantly differ ($F$ = 1.04, $p$ = .35) for people from different backgrounds (Dutch, Western, non-Dutch and non-Western). Also, background did not significantly differ ($\chi^2$ = 3.6, df = 2, $p$ = .2) for participants in paid employment (77.4% Dutch, 12.3% non-Dutch Western, 10.3% non-Dutch non-Western) or not in paid employment (82.5% Dutch, 7.5% non-Dutch Western, 10.0% non-Dutch non-Western).

## Measures

**Socio-economic position (SEP).** Three indicators for SEP were used in the current study; a) highest achieved educational level, b) monthly personal disposable income and c) paid employment (0/1). Highest achieved educational level was measured on a 6-point scale (1 = primary school, 2 = intermediate secondary school/junior high school, 3 = higher secondary education/senior high school, 4 = intermediate vocational education/junior college, 5 = higher vocational education/college, 6 = University). Mean scores were used in the analyses and higher mean scores indicate a higher achieved educational level.

Monthly personal disposable income was asked on a 12-point scale ranging from 0 (no income) to 12 (having an income of > 7500). The mean score of this scale was used in the analyses. A higher mean score indicates a higher monthly personal disposable income.

Paid employment was derived from a question on what the main daily activity of participants was with being in paid employment of one of the answer categories. Being in paid employment was thereafter dichotomized (no/yes).

**Time perspective orientations.** The Short Form of the Zimbardo Time Perspective Inventory [39], derived from the original Zimbardo Time Perspective Inventory [24], was used to measure the five different time perspectives (future, present-fatalistic, present-hedonistic, past-negative, past-positive). The scale consists of 15 items, three items for each subscale. All scales use a five-point Likert-scale response format that ranges from 1 = "*very uncharacteristic*" to 5 = "*very characteristic*". Since no Dutch version of this measure existed yet, a forward and backward translation was performed (English to Dutch). Examples of items are: "Since whatever will be will be, it doesn't really matter what I do" (present-fatalistic) and "It is important to put excitement in my life" (present-hedonism). Only the subscales measuring future (Cronbach's $\alpha$ = .63) and past-negative (Cronbach's $\alpha$ = .85) time perspective showed a sufficient Cronbach's $\alpha$ (> .60). From each of the remaining subscales, one item was deleted resulting in correlations between the two remaining items in each subscale of .43, .63 and .63, for present-fatalistic (item "Life today is too complicated; I would prefer the simpler life of the past" was deleted), past-positive (item "I enjoy stories about how things used to be in the 'good old times'" was deleted) and present-hedonistic time perspective (item "I make decisions on the spur of the moment" was deleted), respectively. Mean scores of each subscale of the five different time perspectives were used in the analyses. Higher mean scores indicate higher scores on the subscales.

**Psychological Capital (PsyCap).** The Compound Psychological Capital Scale (CPC-12) [40] was used to measure PsyCap (full scale Cronbach's $\alpha$ = .89). Again, no Dutch version existed so a forward and backward translation was performed (English to Dutch). This measure consists of twelve items measuring four subcomponents (hope, optimism, resilience, and self-efficacy) using a 6-point likert-scale ranging from 1 = "*completely disagree*" to 6 = "*completely agree*". Examples of items are: "If I am at a dead end or stuck, I could think of

many ways to get out of it (hope)" and "I can solve most problems if I invest the necessary effort (self-efficacy)". Explicit instructions to participants for answering the questions with their own health in mind were included. Previous research has shown that the CPC-12 is a reliable measure for psychological capital, with sufficient convergent and discriminant validity and that it can be used in different life domains [40]. Mean scores on the overall PsyCap construct were used in the analyses, with higher scores indicating a higher PsyCap.

**Financial self-reliance.** Financial self-reliance was measured using responses to the item "To what extent do you have trouble with your money matters, such as paying bills and keeping eye on expenditure?"; ranging from 1 = "*no trouble at all, I do not need help*" to 4 = "*A lot of trouble, I can only realize it with help*". The mean score of the reversed recoding of this item was used in the analyses to indicate a person's financial self-reliance.

**Perceived life stress.** Perceived life stress was measured using the Dutch translation of the Perceived Stress Scale 10 [PSS-10; 41] consisting of 10 items using a 5-point likert scale ranging from 0 = "*never*" to 4 = "*very often*". Examples of items are "In the last month, how often have you felt that you were unable to control the important things in your life?" and "In the last month, how often have you felt confident about your ability to handle your personal problems?". One item was deleted (i.e., 'In the last month, how often have you been able to control irritations in your life?') due to an ambiguity in the Dutch translation. The remaining 9 items showed an adequate internal reliability (Cronbach's $\alpha$ = .86). Mean scores were included in the analyses, with higher scores indicated more perceived life stress.

**Self-reported physical health conditions.** Physical health conditions were measured using 8 dichotomous items asking participants whether they experience a variety of physical health conditions regularly (*no/yes*), including joint pains, heart complaints or chest pain when performing physical effort, headaches, gastro-intestinal problems, respiratory infections or conditions (i.e., shortness of breath and having a cold (coughing and/or having a stuffy nose)), sleep disturbances and fatigue. Summing these items created a variable which indicated the number of different health conditions participants experienced regularly (Cronbach's $\alpha$ = .72) ranging from 0 to 8, with higher scores indicating higher self-reports on experiencing more physical health conditions.

**Self-perceived health.** Self-perceived health was measured using self-reported responses to the item "How would you rate your health in general?"; ranging from 1 = "*bad*" to 5 = "*excellent*".

## Data analysis

Descriptive results were analyzed using SPSS version 25 [42]. Bivariate correlations between the variables under investigation were calculated, variations in proportions were assessed using cross-tabulations with $\chi^2$ tests, and analyses of variances were performed using F-tests to assess differences in means. Structural equation modeling was used, using Mplus7 [43], to estimate the proposed model (see Fig 2). The proposed model was tested using Full information maximum likelihood (FIML) and bootstrapping using maximum likelihood estimation. In all analyses, and at all levels in the model, sex (Men = 1; Women = 2), age and background (included as two dummy variables: a) Western non-Dutch vs. non-western and Dutch; b) non-Western non-Dutch vs. Western and Dutch) were included as covariates.

## Results

### Descriptive results

As expected in hypothesis 1, a lower achieved educational level (SEP indicator) was significantly associated with lower self-perceived health (*Spearman's rho* = .32, $p < .01$; see Table 1) and higher physical health conditions (*Spearman's rho* = -.25, $p < .01$). Compared with people

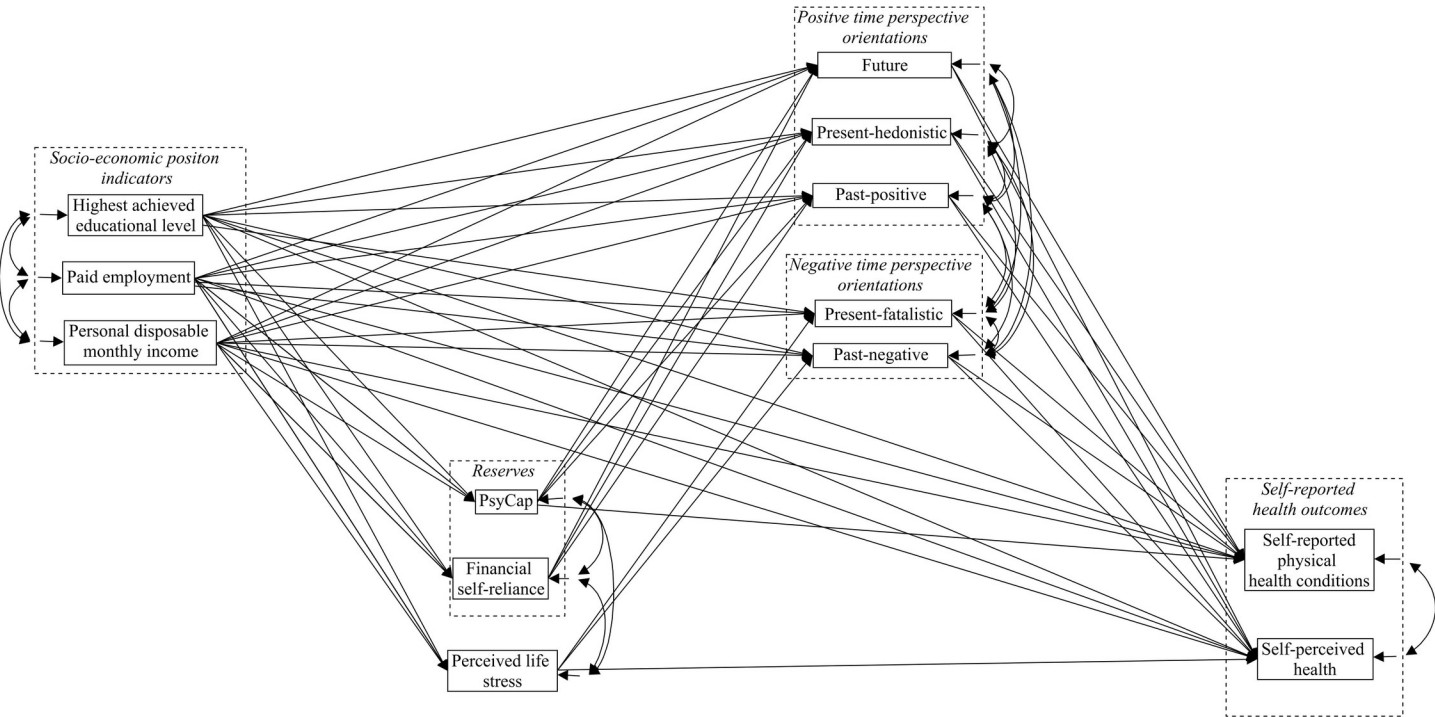

**Fig 2. Conceptual model tested in the current study.** PsyCap = Psychological capital; Although sex, age and background were included as covariates in the analysis at all model levels, estimated paths regarding age, sex and background are not depicted in the Figure.

in paid employment, individuals not in paid employment reported a significant lower self-perceived health (M = 2.87 vs. M = 3.20, respectively; $t$ = -4.61, $p < .001$) and significantly higher levels of physical health conditions (M = 2.18 vs. M = 1.35; $t$ = 5.56, $p < .001$; see Table 2). Furthermore, and in line with hypothesis 1, correlations between model variables (Table 1) showed that a higher personal disposable income (SEP indicator) was significantly associated

**Table 1. Pearson and Spearman's rho correlations between model variables.**

| Measure | 1. | 2. | 3. | 4. | 5. | 6. | 7. | 8. | 9. | 10. | 11. | 12. |
|---|---|---|---|---|---|---|---|---|---|---|---|---|
| 1. Highest achieved educational level[a] | 1 | | | | | | | | | | | |
| 2. Personal disposable monthly income | .50** | 1 | | | | | | | | | | |
| 3. Self-reported physical health conditions | -.25** | -.14** | 1 | | | | | | | | | |
| 4. Self-perceived health | .32** | .15** | -.55** | 1 | | | | | | | | |
| 5. Perceived life stress | -.14** | -.19** | .41** | -.43** | 1 | | | | | | | |
| 6. Psychological capital | .27** | .20** | -.36** | .42** | -.57** | 1 | | | | | | |
| 7. Financial self-reliance | .19** | .15** | -.24** | .19** | -.23** | .17** | 1 | | | | | |
| 8. Past-positive | .06 | .09* | -.17** | .20** | -.27** | .34** | .19** | 1 | | | | |
| 9. Future | .09 | .09* | -.13** | .19** | -.24** | .36** | .16** | .70** | 1 | | | |
| 10. Past-negative | -.18** | -.19** | .33** | -.31** | .48** | -.31** | -.22** | -.26** | -.19** | 1 | | |
| 11. Present-fatalistic | -.23** | -.19** | .12* | -.20** | .09 | -.04 | -.04 | -.01 | -.04 | .21** | 1 | |
| 12. Present-hedonistic | .06 | .01 | -.10* | .13** | -.14** | .31** | .01 | .16** | .20** | .06 | .07 | 1 |

* $p < .05$ (2-tailed)

** $p < .01$ level (2-tailed)

[a] Since highest achieved educational level was measured at a 6-point ordinal scale Spearman's rho was calculated.

**Table 2. Mean scores (SD, mean std. errors) of model variables for paid employment (SEP indicator).**

| | Paid employment | |
|---|---|---|
| | **No** | **Yes** |
| Self-perceived health | 2.87*** (.82, .04) | 3.20 (.70, .05) |
| Range 1–5 | N = 355 | N = 172 |
| Self-reported physical health conditions | 2.18*** (1.99, .11) | 1.35 (1.36, .10) |
| Range 0–8 | N = 355 | N = 172 |
| Perceived life stress | 1.63 (.59, .03) | 1.57 (.55, .05) |
| Range 0–4 | N = 326 | N = 153 |
| Psychological capital | 4.20*** (.81, .05) | 4.49 (.74, .06) |
| Range 1–6 | N = 326 | N = 153 |
| Financial self-reliance | 3.77** (.62, .03) | 3.91 (.38, .03) |
| Range 1–5 | N = 348 | N = 170) |
| Present-hedonistic time perspective | 3.06 (.86, .05) | 3.12 (.84, .07) |
| Range 1–5 | N = 326 | N = 152 |
| Present-fatalistic time perspective | 3.11* (.84, .05) | 2.94 (.85, .07) |
| Range = 1–5 | N = 326 | N = 152 |
| Past-negative time perspective | 2.40** (1.13, .07) | 2.10 (.97, .08) |
| Range 1–5 | N = 326 | N = 152 |
| Past-positive time perspective | 3.89 (.81, 05) | 3.93 (.83, .07) |
| Range 1–5 | N = 326 | N = 152 |
| Future time perspective | 3.86 (.69, .04) | 3.92 (.60, .05) |
| Range 1–5 | N = 326 | N = 152 |
| Personal monthly disposable income | 2.28** (1.79, .09) | 4.26 (1.94, .14) |
| Range 0–11 | N = 400 | N = 197 |

SEP = socio-economic position

* p < .05

** p < .01

*** p < .001.

with lower self-reports on physical health conditions ($r$ = -.14, $p$ < .01) and significantly higher self-perceived health ($r$ = .15, $p$ < .01).

As in line with hypothesis 2a, a higher achieved educational level was significantly associated with higher reported PsyCap score (*Spearman's rho* = .27, $p$ < .01) and higher self-reports on financial self-reliance (*Spearman's rho* = .19, $p$ < .01). People not in paid employment, compared with people in paid employment showed, as expected, a significantly lower mean PsyCap score (M = 4.20 vs. M = 4.49, $t$ = -3.77, $p$ < .001) and lower mean financial self-reliance score (M = 3.77 vs. M = 3.91, $t$ = -2.99, $p$ < .01). Furthermore, correlations between model variables (Table 1) confirmed a significant positive association between personal disposable income (SEP indicator) and the reserves PsyCap ($r$ = .20, $p$ < .01) and financial self-reliance ($r$ = .15, $p$ < .01).

The descriptive results showed that two of the three measured SEP indicators were significantly associated with perceived life stress, which was partly in line with expectations of hypothesis 2b. As expected, a negative association between highest achieved educational level and perceived life stress (*Spearman's rho* = -.14, $p$ < .01) and between personal disposable monthly income and perceived life stress (*Spearman's rho* = -.19, $p$ < .01) was found. On the other hand, no significant difference was found in mean scores of perceived life stress between people in employment (Table 2).

Furthermore, hypothesis 2c was also only partly confirmed. Correlations between model variables showed, as expected, significant positive associations between personal disposable monthly income and positive time perspective orientations (past-positive: $r$ = .09, $p$ < .05); future: $r$ = .09, $p$ < .05). Nevertheless, no association was found between personal disposable monthly income and a present hedonistic time perspective. Also, no significant associations between highest achieved educational level, being in paid employment and positive time perspective orientations were found (Tables 1 and 2).

In line with hypothesis 2d, highest achieved educational level was significantly associated with negative time perspective orientations, indicating that a higher highest achieved educational level was associated with significantly lower scores on present fatalistic (*Spearman's rho* = -.23, $p$ < .01) and past negative time perspective (*Spearman's rho* = -.18, $p$ < .01). Also, a significant negative association was found between personal disposable income and a past-negative and present-fatalistic time perspective ($r$ = -.19 and $r$ = -.19, respectively, $p$ < .01). People not in paid employment compared with in paid employment showed a significantly higher present-fatalistic (M = 3.11 vs. M = 2.94, respectively; $t$ = 2.00, $p$ < .05) and past-negative time perspective (M = 2.40 vs. M = 2.10, respectively, $t$ = 3.01, $p$ < .01) mean score.

In line with hypothesis 3a, the reserves PsyCap and financial self-reliance were positively associated with better self-reported health outcomes (higher self-perceived health and lower self-reported physical health conditions; hypothesis 3a, see Table 1). On the other hand, and as expected in hypothesis 3b, perceived life stress was positively associated with worse self-reported health outcomes (higher self-reported physical health conditions and lower self-perceived health).

Furthermore, the reserves PsyCap and financial self-reliance showed to be positively associated with higher scores on future and past-positive time perspective orientations (hypothesis 4a). Regarding a present-hedonistic time perspective, only for the intrapersonal reserve PsyCap a positive association was found (hypothesis 4a).

Confirming partly hypothesis 4b, perceived life-stress showed a positive association with a higher past-negative time perspective. Nevertheless, no association was found between perceived life-stress and a present fatalistic time perspective.

Higher self-reports of positive time perspective orientations showed to be positively associated with increased self-reported health outcomes (fewer self-reported physical health conditions and higher self-perceived health; hypothesis 5a), whereas higher self-reports of negative time perspectives showed to be associated with worse self-reported health outcomes (more self-reported physical health conditions and lower self-perceived health; hypothesis 5b).

## Structural equation modeling

An overview of the significant model results of the structural equation model regarding self-perceived health and self-reported physical health conditions is presented in Fig 3. The tested model showed a good model fit (RMSEA = .031; SRMR = .011; CFI = .997). Results showed that the model explained 32.9% (p = .000) of the variance in self-perceived health, and 28.7% (p = .000) of the variance in self-reported physical health conditions. Two of the three tested SEP indicators showed a significant association with the self-reported health outcomes. A higher achieved educational level showed to be significantly associated with lower self-reports of physical health conditions (std. estimate = -.12; unstd. 95% CI [-.25;-.04]) and higher self-reports of self-perceived health (std. estimate = .12; unstd. 95% CI [.02; .11]). Also, being in paid employment showed to be significantly associated with lower self-reported physical health conditions (std. estimate = -.10; unstd. 95% CI [-.72;-.05]). Personal disposable monthly income showed not to be associated with the self-reported health outcome measures. These results partly confirm hypothesis 1; lower SEP indicators are associated with higher self-reports

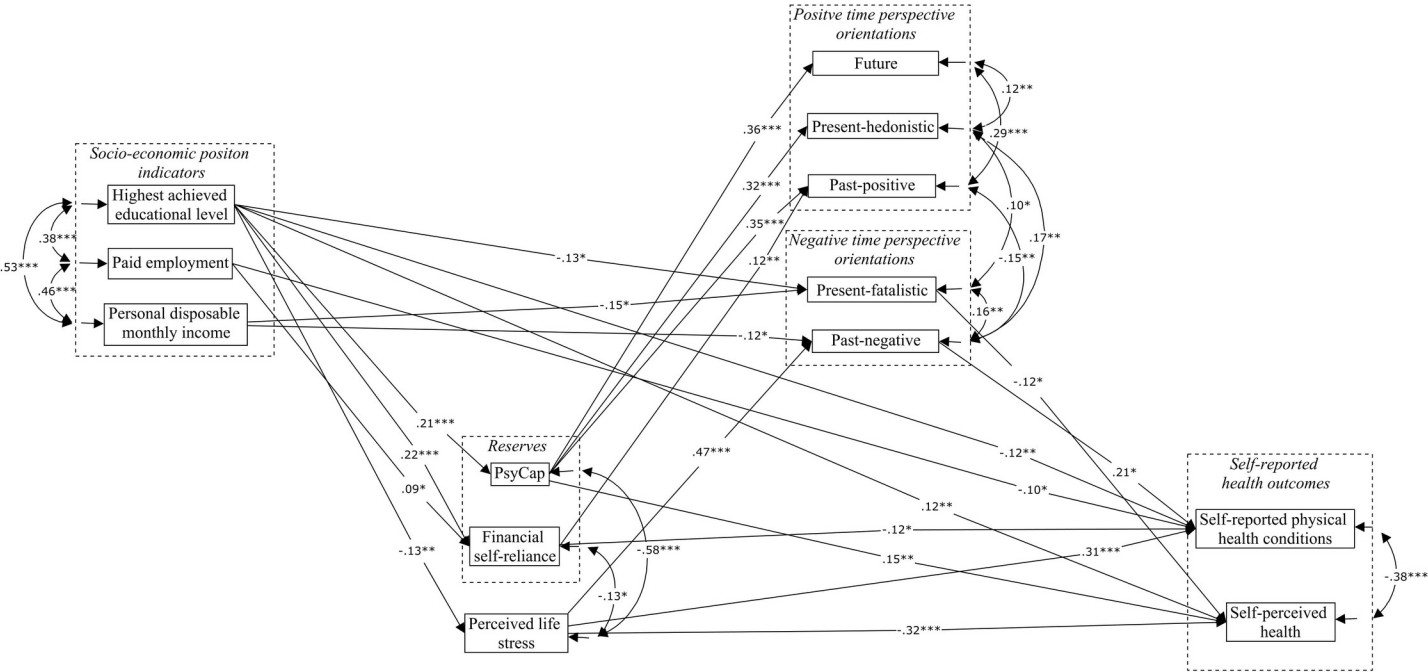

**Fig 3. Significant model results.** * p < .05; ** p < .01; *** p < .001; RMSEA = .031; SRMR = .011; CFI = .997; PsyCap = Psychological capital; Although sex, age and background were included as covariates in the analysis at all model levels, model results regarding age, sex and background are not depicted in the Figure. Standardized (YX) significant results are displayed for all other model variables; Significant mediation effects were found for the following paths: H6a: Highest achieved educational level → PsyCap → self-perceived health; Highest achieved educational level → financial self-reliance → self-reported physical health conditions; H6b: Highest achieved educational level → perceived life stress → self-perceived health; Highest achieved educational level → perceived life stress → self-reported physical health conditions.

of physical health conditions and lower self-reports of self-perceived health. Also, significant associations were found between two of the three tested SEP indicators and intrapersonal and tangible reserves, confirming hypothesis 2a. A significant association of highest achieved educational level was found with self-reported PsyCap (std. estimate = .21; unstd. 95% CI [.06; .16]) and with financial self-reliance (std. estimate = .22; unstd.95% CI [.04;.12]). Also, a significant association between paid employment and financial self-reliance was found (std. estimate = .09; unstd. 95% CI [.01;.22]).

In line with hypothesis 2b, a negative significant association was found between highest achieved educational level and perceived life stress (std. estimate = -.13; unstd. 95% CI [-.09;-.02]). Nevertheless, no significant association was found between paid employment or personal disposable monthly income and perceived life stress.

Moreover, hypotheses 2c was, not confirmed by the results; none of the tested SEP indicators were associated with positive time perspective associations when controlling for all covariates. On the other hand, hypothesis 2d was, for the main part, confirmed by the structural equation model results; indicators of a lower SEP were associated with higher scores on negative time perspective orientations. Highest achieved educational level (std. estimate = -.13; unstd. 95% CI [-.13;-.03]) and personal disposable monthly income (std. estimate = -.15; unstd. 95% CI [-.11;-.01]) were negatively associated with a present fatalistic time perspective. Personal disposable monthly income was also negatively (std. estimate = -.12; unstd. 95% CI [-.12;-.01]) associated with a past-negative time perspective.

Results were also in line with hypotheses 3a and 3b; a significant positive association of self-reported PsyCap (std. estimate = .15; unstd. 95% CI [.05; .26]) and a significant negative association of perceived life stress (std. estimate = -.32; unstd. 95% CI [-.60; -.28]) was found with

self-perceived health. Regarding self-reported physical health conditions, a significant positive association with perceived life stress was found (std. estimate = .31 (unstd. 95% CI [.65; 1.31]). and a significant negative association with financial self-reliance (std. estimate = -.12; unstd. 95% CI [-.70; -.09]). Furthermore, the structural equation model result confirmed, for the most part, hypothesis 4a and 4b. PsyCap was indeed significantly associated with higher self-reports of positive times perspective orientations (past-positive, future, and present-hedonistic; standardized estimates were .35 (unstd. 95% CI .26; .45), .36 (unstd. 95% CI [.22; .38]), and .32 (unstd. 95% CI.[25; .44]), respectively) and financial self-reliance with a past-positive time perspective (std. estimate = .12; unstd. 95% CI [.04; .30]). On the other hand, and as expected by hypothesis 4b, perceived life stress was significantly associated with higher self-reports of the past-negative time perspective (std. estimate = .47; unstd. 95% CI [.71; 1.05]). Nevertheless, contrary to what was expected in hypothesis 4b, perceived life stress was not significantly associated with higher self-reports of a present-fatalistic time perspective, after controlling for SEP indicators, age, sex and background. In addition, hypothesis 5a was not confirmed by the structural equation model results; positive time perspective orientations did not have any association with self-reported health outcomes when controlling for all SEP indicators, reserves, perceived life stress and negative time perspectives. On the other hand, higher rates of the negative time perspective orientations showed to be associated with worse self-reported health outcomes, confirming hypothesis 5b. A present-fatalistic time perspective showed a significant negative association with self-perceived health (Std. estimate = -.12; unstd. 95% CI [-.17;-.02]) and a past-negative time perspective showed a significant positive association (Std. estimate = .21; unstd. 95% CI [.03; .40]) with self-reported physical health conditions.

Moreover, significant findings were found for three single mediation paths. The association of the SEP indicator highest achieved educational level with self-perceived health showed to be significantly mediated by PsyCap (std. estimate = .03; 95% CI [.01;.05]), and perceived life stress (std. estimate = .04; 95% CI [.01; .07]), not by financial self-reliance. The association of a higher achieved educational level with lower self-reported physical health conditions was significantly mediated by lower perceived life stress (std. estimate = -.04; 95% CI [-.07;-.01]) and by higher financial self-reliance (std. estimate = -.03; 95% CI [-.04;-.01]), not by lower self-reported PsyCap. These results confirm most of the expectations in hypothesis 6a and 6b. No significant results were found regarding the single mediation paths through the three positive and two negative time perspective orientations (hypothesis 6c and 6d). Furthermore, regarding the double mediation paths through either a) the reserves and positive time perspective orientations (hypothesis 7a) or b) perceived life stress and negative time perspective orientations (hypothesis 7b), no significant results were found.

## Discussion

The main aim of the current study was to gain insight into the role of several psychological variables in the SEP-Health gradient. More specifically, our aim was to investigate whether lower self-perceived health and higher self-reported physical health conditions among individuals reporting lower levels on SEP indicators can be explained by lower self-reports of intrapersonal (PsyCap) and tangible (financial self-reliance) reserves, higher perceived levels of life stress, and, in turn, more negative (or less positive) self-reports on time perspective orientations. Clarifying these possible explanatory pathways in the SEP-health gradient can provide input for the design of timely targeted prevention and intervention strategies. Although not all our hypotheses were confirmed, the results were largely in line with our expectations, and provide further support for the RCM.

First, our results showed that highest achieved educational level and being in paid employment seem to play a larger role in the social stratification within self-reported and self-perceived health outcomes than personal disposable income. Controlling for all other model variables, personal disposable monthly income did not show an association with self-reports on physical health conditions and self-perceived health (only an association with negative time perspective orientations).

Secondly, although there are significant and expected mediating paths from SEP indicators to both self-reported health-related outcomes, this effect was only found for the SEP indicator highest achieved educational level. Being in paid employment did show a significant association with a higher self-perceived health when controlling for all other model variables, nevertheless, this association was not mediated by the included mediators in the model. Furthermore, a direct effect of highest achieved educational level (SEP indicator) on self-reported health conditions and self-perceived health remained significant after including the mediators in the analyses and controlling for several other important covariates. This is in line with previous research in European countries [e.g., 10, 44], and it suggests that the current model does not fully capture the underlying mechanisms explaining the SEP-health gradient. Our results do indicate that perceived life stress, and reserves (PsyCap and financial self-reliance) play a significant (mediating) role in explaining why individuals with a lower achieved educational level (SEP indicator) show lower self-perceived health and higher self-reports of physical health conditions, compared with individuals with a higher achieved educational level. Our results thus suggest that the higher experienced stress and lower reserves (psychological capital–hope, optimism, resilience and efficacy–and financial self-reliance) could make individuals with a lower SEP more vulnerable for perceived health conditions or perceiving their health as poor. Indeed, the negative effect of a lack of optimism on the frequency and severity of illness was already established by Scioli et al. [45], and more recently, Krasikova, Lester, and Harms [46] established that high levels of psychological capital were positively related to self-perceived health and substance use among soldiers. The current findings thus also suggest that PsyCap could have a buffering effect in the negative effects of stress (and SEP) on self-reported health outcomes [see also 47], and that by increasing PsyCap, one could alleviate some of the negative relationships between these variables.

Our expectations regarding the influence of time perspective orientations on self-reported health conditions and self-perceived health were only partly confirmed: In line with previous research [e.g., 48], the negative time perspective orientations (present-fatalistic and past-negative) were associated with worse self-reported health outcomes. Also, reserves (PsyCap and/or financial self-reliance) were positively associated with positive time perspective orientations, whereas perceived life stress was positively associated with a past-negative time perspective. Although a significant association was found between SEP indicators indicating a lower SEP (lower achieved educational level and lower personal disposable income) and negative time perspective orientations, the SEP-health gradient was, in contrast to our expectations, not explained (mediated) by negative time perspective orientations. As Adams and White [49] note, the lack of associations between future-thinking and self-reported health outcomes could stem from the context of the Netherlands as a 'welfare' state, which is likely to affect individuals' thinking about future uncertain events and their impact on their health. Specifically, the universal access to (relatively) low-cost healthcare in the country could provide a sense of security that future health issues will be resolved as well, and thus not affect current-day self-perceived health. Further, we did not take specific time horizons into account: There are indications that individuals differ in their perception of 'future', and that health behaviour is affected by these time horizons, and vice versa. For example, Petry, Bickel, and Arnett [50] reported that heroin addicts conceptualized 'future' in terms of the very near future (e.g., 1

hour away), whereas healthy controls conceptualized the future as 7 days away. Future research should therefore also measure these time horizons in participants, which in turn could influence psychological time perspective.

## Practical implications resulting from this study

The results from this study provide starting points for the design of health prevention and promotion efforts. Specifically, our results suggest that the SEP-Health gradient can be reduced by focussing on–preferably simultaneously–increasing PsyCap and financial self-reliance and increased coping with the perceived levels of life stress for individuals with a low achieved educational level, since these mediation paths showed the strongest effects. The current research also suggests that interventions efforts should be developed specifically and only for individuals with a low achieved educational level, since targeting both high and low educated groups with the same intervention is likely to inadvertently increase the health gradient [e.g., 51].

Previous research using randomized controlled designs has shown that PsyCap can be significantly increased by brief (2–3 hours), one-session, interactive micro-interventions, showing effects directly after the intervention as well as after one month [52] and after eight weeks [53]. These effects are also found among lower SEP individuals, for instance among homeless women [22]. PsyCap interventions are typically conducted in small groups and consist of exercises, using self- and group-reflection, focusing on discussing how to set realistic and specific (health) goals and how to plan for and overcome potential obstacle on the way to attaining the goal [54]. Furthermore, the group discussions are supervised by facilitators who encourage positive self-talk, positive thinking and vicarious learning among participants through peer role modelling, which, in turn, additionally could increase participants' intrapersonal reserves in the form of social support. Moreover, offering PsyCap interventions to individuals with a low SEP, for instance by offering such an intervention at workplaces where usually many people with a low SEP work, could therefore be useful to reduce the SEP-health gradient.

With respect to perceived life stress, interventions aimed at increasing stress management or coping skills, such as using relaxation techniques, guided imagery, coping self-statements, and/or mindfulness-based stress reduction are recommended. There are indications for the effectiveness of such interventions, for example, a randomized controlled trial investigating the effectiveness of a mindfulness-based stress reduction intervention among individuals diagnosed with Generalized Anxiety Disorder, has shown that such an intervention improved stress reactivity and coping measured in a laboratory stress challenge [55]. Furthermore, research among low-income Mexican American women showed that social support buffered the negative effect of stress on mental health, specifically postpartum depression [56]. Thus, it is worthwhile to design interventions that target not only coping, but also incorporate an element of social support to reduce the negative impact of stress on health.

Nevertheless, although social stratification is associated with differences in perceived stress and psycho-social reserves, we should not be blind to the fact that social stratification is also a characteristic of a society in which inequities in power, money and resources are the structural drivers of health inequities [2]. The socioeconomic and political context is important in explaining health inequities. We should therefore bear in mind that individual indicators of SEP are derived from social and economic processes that shape the distribution of education, occupations and income across the population. The fact that, in the Netherlands, teachers have lower expectations of children with a low SEP, resulting in lower support from primary and secondary schools for their attempts to access tertiary education even when they have satisfactory test scores [14], is a serious cause for concern. In order to decrease the SEP-Health gap, in line with our results, especially teachers and schools should be educated to decrease this stigma for children with a low SEP to

narrow this growing opportunity gap in the Netherlands. Furthermore, results of our study indicate that to decrease self-reported physical health conditions, political action is needed to increase employment, especially under people with a low SEP. Future research should also focus on differences in regions or countries where variety in these macro factors can be investigated that may further elucidate the mechanisms behind the SEP-Health gradient.

## Strengths, limitations and implications for future research

Although we conducted our research in a representative general Dutch population sample, which enables the generalizability of the findings to the general Dutch population, limitations need to be acknowledged. First, it is important to state that individuals scoring lower on SEP indicators were oversampled in our study to ensure sufficient variability in the data collected. Thus, although our results give useful insights into the relative health disadvantage of people scoring lower on the measured SEP indicators, our results do not elucidate the public health importance of the socioeconomic health inequality in terms of the size of the exposed population or absolute level of risk.

Secondly, a broad age range (16–96 years) was used in the current study, nevertheless, SEP indicators and time perspectives may change or develop over time in a persons' life and therefore may have a different effect in a different stage of their lives [57]. Also, no separate analyses for men and women were performed, nevertheless men and women may deal with stress in a different way or may think differently about the future. We did control for sex and age at every level of our model to minimize the effect this could have had on our results. Future research should use a bigger sample to be able to conduct age and gender specific group analyses to gain insight into defining more specific implications for health promotion, prevention and management for different subgroups.

Furthermore, although previous research as well as theory strongly supports the directional hypotheses of our model, this study used a cross-sectional design, and therefore no conclusions about causal effects can be drawn.

The questionnaire used to measure time perspective, although validated in a previous study [39], showed rather weak internal consistencies for several of the subscales. This might have resulted in finding less strong (or null) associations with other model variables. Future studies should therefore use longitudinal designs (also including future health) and include a different measure for time perspective with better psychometric properties. For example, the Consideration of Future Consequences Scale [26] could be employed to measure future thinking, and trait or state impulsivity scales could be used as a proxy for present-focused thinking (and future discounting). Further, as mentioned above, it could be worthwhile to measure specific time horizons to gain insight into the concept of 'future' for individuals low or high in SEP.

Although this study included a more comprehensive model than most previously conducted studies [e.g., 35] such that more associations proposed by the reserve capacity model could be tested, a limitation is that the current study only included one intrapersonal and tangible reserve, i.e., PsyCap and financial self-reliance, respectively. Future research should also include relational reserves, such as perceived social support, to gain more insight into possible protective factors which, for instance, could buffer the negative impact of perceived stress on health. Moreover, the effect of a low achieved educational level on the self-reported health outcomes used in the current study was only partly mediated by lower levels of PsyCap, financial self-reliance and higher perceptions of life stress. Therefore, more research is needed to gain insight into other factors that can explain why low SEP indicators result in worse self-perceived health outcomes. For instance, previous research has indicated health literacy as part of the mechanism explaining the relationship between education and health [58].

Moreover, this study explicitly looked at subjectively measured health outcome measures. Although self-rated or perceived health covers a variety of health outcomes and has shown to be consistently associated with morbidity and mortality [6, 7], the mechanisms explaining health inequalities regarding more objective measured health outcomes might be different. Future research on elucidating the mechanisms behind health inequalities should therefore also include more objectively measured health outcomes.

## Conclusions

To conclude, this study successfully extents current scientific literature by providing more insight into the role psychological variables play in the SEP-Health gradient. Psychological capital (hope, efficacy, resilience, optimism), financial self-reliance and perceived life stress showed to mediate the association between a lower achieved educational level and self-reported health outcomes and therefore play a large role in explaining this association. Furthermore, this study provides practical knowledge to develop timely prevention efforts aimed at reducing the SEP-Health gradient. Prevention efforts trying to reduce the SEP-Health gradient should focus on simultaneously increasing PsyCap and financial self-reliance and increasing coping levels with perceived life stress for individuals with a low achieved educational level. Moreover, prevention efforts should additionally focus on narrowing the opportunity gap in education between children with a low versus a higher SEP (e.g., by raising awareness of teachers and schools regarding SEP-related stigmatization of their students) and reduce unemployment among people with a low SEP.

## Author Contributions

**Conceptualization:** Karen Schelleman-Offermans, Karlijn Massar.

**Formal analysis:** Karen Schelleman-Offermans.

**Funding acquisition:** Karlijn Massar.

**Methodology:** Karen Schelleman-Offermans.

**Supervision:** Karlijn Massar.

**Writing – original draft:** Karen Schelleman-Offermans.

**Writing – review & editing:** Karlijn Massar.

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
