## [Decision Letter · Decision Letter 0]

28 May 2020

PONE-D-20-03476

Explaining socioeconomic inequalities in self-reported health: The mediating role of perceived life stress, psychological capital, and time perspective.

PLOS ONE

Dear Dr. Schelleman-Offermans,

Thank you for submitting your manuscript to PLOS ONE. After careful consideration, we feel that it has merit but does not fully meet PLOS ONE’s publication criteria as it currently stands. Therefore, we invite you to submit a revised version of the manuscript that addresses the points raised during the review process.

You will find 5 thoughtful reviews from experts in the field. While I feel that your revision should address all of the comments and concerns raised by the reviewers, I would encourage to focus particularly on the following issues:

As noted by reviewer 3: please make sure that your introduction is clear with regards to rationalizing and providing more information about your dependent measure. This should include a more thorough discussion of Marmot's work and its relation to this study.

This reviewer notes that it is also important to make very clear the distinction between perceptions/reality.

Reviewer 4 also notes that the hypotheses and review of the literature are not well integrated. This reviewer also notes that it is important to note earlier on in the manuscript that the data is from the Nertherlands and to situate the data within the context of the the Netherlands. 

Comment 1 from reviewer 5 should also be given special attention. They raise a concern about integration of Marmot's work in the introduction.

Reviewer 5 also suggests a more thorough consideration of implications, see comment 9.

We look forward to receiving your revised manuscript.

Kind regards,

Neha John-Henderson

Academic Editor

PLOS ONE

Journal Requirements:

2. Please include additional information regarding the survey or questionnaire used in the study and ensure that you have provided sufficient details that others could replicate the analyses. For instance, if you developed a questionnaire as part of this study and it is not under a copyright more restrictive than CC-BY, please include a copy, in both the original language and English, as Supporting Information. Moreover, please include more details on how the questionnaire was pre-tested, and whether it was validated.

Reviewers' comments:

Reviewer's Responses to Questions

**Comments to the Author**

1. Is the manuscript technically sound, and do the data support the conclusions?

Reviewer #1: No

Reviewer #2: Yes

Reviewer #3: Yes

Reviewer #4: No

Reviewer #5: Partly

2. Has the statistical analysis been performed appropriately and rigorously? 

Reviewer #1: N/A

Reviewer #2: Yes

Reviewer #3: Yes

Reviewer #4: No

Reviewer #5: Yes

3. Have the authors made all data underlying the findings in their manuscript fully available?

Reviewer #1: Yes

Reviewer #2: Yes

Reviewer #3: Yes

Reviewer #4: Yes

Reviewer #5: Yes

4. Is the manuscript presented in an intelligible fashion and written in standard English?

Reviewer #1: Yes

Reviewer #2: Yes

Reviewer #3: No

Reviewer #4: Yes

Reviewer #5: Yes

5. Review Comments to the Author

Reviewer #1: This study has some interest in analyzing the mediating role of psychological factors in the relationship between SocioEconomic position (SEP) and self-rated health.

However, the methods and the data do not support the conclusion: «Psychological capital (hope, efficacy, resilience, optimism) and perceived life stress showed to mediate the SEP-Health gradient and therefore play a large role in explaining it (543-545). There is a lack of coherence between the theory and the methods resulting in the omission of social variables, which paves the way for a psychological explanation of the relationship under study.

The premise of the author’s argumentation is that resources to change structural factors through policies (education, income, migration status, etc.) requests too many resources, so there is a need to concentrate on individuals and psychological factors. Is there a good reason to elude fundamental factors that are well known and demonstrated in the literature on social inequities in health (please see the commission on the social determinants of health): living and working conditions, social exclusion, migration status, racism and other discriminations, access to health care, etc.

The approach of the authors reduce social inequalities to psychological inequalities between individuals. In addition to this too individualistic approach, it has been shown that groups are affected by social inequalities. Furthermore the social gradient in health demonstrate that inequalities are relative to the position in the social structure, that related to power relationship. Please see social epidemiology (Lynch and Kaplan, Social position and health, in Berkman and Kawachi, 2000)

The concept of ‘reserve capacities’ are defined by authors as composed of three dimensions: « tangible reserves (e.g., money and transportation). intrapersonal reserves (e.g., self-esteem and psychological capital) and interpersonal reserves (e.g., social support).» (59-62). However, authors are declaring the following:

«To sum up then, the literature on the RCM indicates that psycho-cognitive variables, or reserve capacities, play a mediating role in explaining the (indirect) effect of SEP on various health outcomes, and as such illustrates the usefulness of the RCM in understanding the psychosocial pathways leading to socioeconomic health inequalities.(95-98).

The following independents variables are included: SEP, stress, time, psychological capital. But what about interpersonal and tangible reserve? What about migration status, region of birth, living and working conditions, discrimination, social network, and access to health care ?

In fact, the model that is tested is made of psychological model.

Authors state in the discussion that « the results were largely in line with our expectations, and provide further support for the RCM » (lines 444) and that the current model does not fully capture the underlying mechanisms explaining the SEP-health gradient (lines 448-49). Indeed, the RCM was only partly tested (intrapersonal reserves), and the important social factors that are cited above were not taken into account. In most WHO models individual factors are located at the end of the model, after macro and living and working conditions.

I would also disagree with the proposed intervention focusing on individuals. As if, finally, the causes of social inequalities are psychological and the intervention is also psychological.

I do not want to underestimate the psychological dimension of human beings, but the explanation of deleterious effects of socioeconomic inequalities cannot be reduced to individuals’ psychology.

Reviewer #2: Review: Explaining socioeconomic inequalities in self-reported health: The mediating role of perceived life stress, psychological capital, and time perspective. – PONE-D-20-03476.

Comments to the authors

General comment:

This study investigates what role perceived life stress, psychological capital and time perspectives play in explaining socioeconomic inequalities in self-reported perceived health and physical health complaints. Overall, the paper is well written, the data are valuable, the scope of the study is interesting and the SEM analytic strategy is appropriate. This being said, there are some minor issues that need to be addressed.

Specific comments:

1. Authors should avoid the use of terms like “rates” when describing self-reported health outcomes along the manuscript. Similarly, they should not refer to “effects” when testing associations in cross-sectional analysis.

2. Line 174-80. Hypotheses H6a, H6b, H7a and H7b are not shown in Figure 1, as the authors mentioned in the text. Please, correct this.

3. Line 182-183. Please, specify that you refer to a direct association between low SEP with more negative self-reported health outcomes.

4. I assume that for methodological reasons the authors dichotomized the SEP indicator. This involves a loss of information about the intermediate spectrum f the SES gradient. Authors should mention this as a limitation of the study. Further, they should state in the discussion the potential bias in self-reported health outcomes associated with gender and age.

5. Line 260-262. “The Short Form of the Zimbardo Time Perspective Inventory [31], derived from the original Zimbardo Time Perspective Inventory [18], was used to measure the five different time perspectives (present-fatalistic, present-hedonistic, past-negative, past-positive).” Please, note that four time perspectives are indicated.

6. Table 1. Please, do not include “mean” in the description of the variables of Table 1. Mean and SD are already indicated in the columns.

Reviewer #3: This paper has the ambition to extend literature on (perceptions of) health by studying whether

psychological capital and feelings of stress mediate the impact of socio-economic inequality. This is an

interesting and relevant topic to study but unfortunately there are fundamental theoretical and

conceptual flaws throughout the paper. The authors study a good number of relevant associations and

competently test these, but need to provide the same level of clarity in embedding their paper in the

relevant literature and in supporting their conceptual choices. Below, I outline my main comments:

1. In the introduction, when discussing the social gradient, the discussion on what socio-economic

position exactly entails and how it is important is superficial. The authors do no define what

socio-economic position is and overlook the decades of research on social stratification where a

distinction is made between social class, status, income inequality, different levels of capital,…

The authors could start by looking at an older overview by Bergman & Joye (2001) and more

recent work by Weeden, Wodtke, Oesch,…

2. The problem statements is not fully clear until the end of the introduction. In the first few pages,

the authors discuss life expectancy, health behavior, and health inequalities almost as

synonyms. In health sociological and epidemiological research, however, different mechanisms

explain disparities regarding these distinct concepts (see the work of Marmot, Singh-Manoux,

Demakakos,…). Since the dependent variable ultimately seems to be subjective health

perceptions, it is imperative that the authors both clarify this early on and in addition

conceptually disentangle this from other concepts related to health.

3. Arguing that (perceived) health inequalities exist solely because of differences in resources is

too simple and ignores a broad literature on the interplay between health-related choices, (lack

of) resources, cognitive differences, stress,… in understanding health inequalities.

4. Especially problematic throughout the manuscript is the constant confusing of reality and

perceptions thereof. Since the authors study self-reported health and assume a mediation of

psychological capital on the pathway from SEP to self-reported health it is problematic that they

do not pay attention to the distinction between those concepts that are measured in an

objective manner and those that tap into subjective perceptions (of health). An emerging

sociological/epidemiological literature has illustrated that these need to be disentangled since

different mechanisms cause material and subjective differences, while both have a disparate

impact on health outcomes.

5. From line 175, the authors start with introducing hypotheses 6a-7b, while the other hypotheses

are introduced later. While this is a strange order to say the least, the problem here mainly lies

in the fact that the hypotheses often are not logically deducted from the literature. This is due

to the fact that the literature review frequently confuses distinct concepts and/or ignores

important strands of the literature on social inequality and health (perceptions)

6. From line 254 the authors introduce the usage of income and educational background as

indicators of socio-economic position. No theoretical support of this choice is provided while the

introduction does not include any discussion of the literature using such a conceptualization of

socio-economic inequality. Why would income and/or educational background be a valid way of

measuring socio-economic inequality?

Reviewer #4: Thank you for the opportunity of reviewing the manuscript ‘Explaining socioeconomic inequalities in self-reported health: The mediating role of perceived life stress, psychological capital, and time perspective’. The manuscript addresses an important topic but I believe it is currently not suitable for publication in PloOne.

Overall, the strength of the paper lies in the very comprehensive description of the psychological constructs, the mediating model and the discussion of such mediating role. However, in order for the manuscript to make a contribution the authors should integrate what is a very psychologically oriented study within the broader understanding of health disparities derived from sociology, economics or social policy.

This is very apparent in how the authors consider socio-economic position and how they conceive the psychological constructs.

The authors consider differences in self-reported health by socio-economic position and the mediating role of three psychological constructs: stress, psychological capital and time discounting. The conceptualisation of socio-economic position (SEP) in the manuscript is poor. It remains unclear why the authors discuss SEP rather than socio-economic status (SES). Operationally, SEP which is a latent construct is considered coarsely through a high-low conceptualisation based on arbitrary thresholds of disposable net income and educational attainment. This is a crucial flaw that invalidates all the analyses. This methodological problem is compounded by the failure of the author/s to discuss how and why SEP may shape individual health not only indirectly through its role in determining stress, psychological capital and time perspectives but also directly. A really large literature examines socio-economic disparities in health and the authors do not discuss/report any of this. They simply acknowledge in the discussion section that direct ‘residual’ and statistically significant association ‘remains’.

The hypotheses and review of the literature are not integrated. The review of theories and prior studies should be woven in the formulation of testable hypotheses.

We learn only on page 10 that the study is based on data from the Netherlands. Given that the study is about the role socio-economic position plays in shaping health outcomes, it would be important to situate the research within the broader economic, social, health and welfare policies that operate in the Netherlands vs., for example, the United States, the United Kingdom or other countries. Both overall levels of health disparities and which factors shape variability across social groups can be expected to vary depending on, just to name an obvious feature, if health care is provided free of charge to all or depends on employer provided health insurance.

I suggest the authors discuss the characteristics of individuals who were excluded from the sample because of missing information. Why not use multiple imputation to recover them? This would also allow to have a larger sample size.

Reviewer #5: This is an important topic which needs ongoing research given the increasing socioeconomic and health disparities.

There are several areas where the current paper would benefit from changes and improvements.

1. Although the authors cite the work of Michael Marmot as the first two references, it is unclear whether they have fully understood his central arguments or is it the way the study questions are drafted that leaves room for concern. Marmot and many others have repeatedly pointed out that health conditions experienced by people, particularly chronic health conditions are predicated on several risk factors that occur over the lifespan and are often beyond the control of the individuals. In fact, many of the SES [or SEP] inequities stem from systemic macro-level discriminatory policies and programs that often favor those already advantaged over the poor. It would be helpful to state these more clearly and expand on these important set of issues.

2. There is often a tendency for researchers to look at cause-effect relationship using psycho-social variables to assess individual level variability but with little acknowledgement of the larger issues. The current paper falls into the same trap where there is an inherent assumption that “individuals have/can have control over their lives” and “are in a position to improve their health by positive thinking” – be it in the form of Reserve Capacity Model/theory or Psychological Capital. The downside of such assumptions is that it leads to unbalanced discussion of underlying factors that produce health and ill-health. It would be useful for the authors to review the paper and redraft some sections with more open and upfront acknowledgment of the limitations and problematic implications of the theories they have used.

3. It would be useful to bring in some definitional issues about SEP and cut-offs. The authors have used a particular definition (education and income) but without explaining why two metrics were chosen. For an international audience, some context would be helpful.

4. I looked up the LISS data archive but am unable to correlate the time-frame used by the study with the Waves listed in the LISS Data Archive. It would be useful to include some information on this aspect.

5. Although the study has controlled for age in the final analyses, but the biological issues of a huge age range of 16-92 remains problematic. Very few young people (<30) in lower SEP would have health conditions that lead to poor health rating and their outlook would be generally positive. On the other hand, many older people would experience a number of chronic health conditions over the lifespan and for a 70 or 80-year old these issues get compounded and confounded. A more nuanced description in the Introduction with relevant references and a more balanced set of statements in the Discussion section would be of value.

6. Perhaps it would be useful to use some right censoring to limit very old age groups to limit the confounding caused by increased frailty and multimorbidity. Whilst this may sound “ageist”, it makes little sense from a population health perspective to include such a large age range as the implications for action are very different in terms of health promotion, prevention and management. I realize this is not the central focus of the paper but this the reality of individuals in the study sample; and has significant implications for population level health issues.

7. Sampling. The authors outline stratified random sampling based on age, gender [sex] with oversampling of LISS sample for low SEP. It is not clear how this then remains a random sample and what adjustment were made in the analyses to adjust for oversampling. These issues are important if population level inferences are to be drawn from the study.

8. I find use of the term “health complaint” highly problematic. Perhaps it is disciplinary differences but when respondents have been asked about health issues many of which are almost diagnostic categories [except for headache and fatigue], then a more appropriate term would be health conditions. These are not psychosomatic problems but actual health conditions. Reviewing LISS variables they are listed as health problems but include a combination of the both categories. A few short explanatory sentences should suffice.

9. With regards to Implications, I was somewhat disappointed that there is no mention of improving health literacy. Whilst macro-level change to reduce inequities [SEP] are harder for researchers to influence, effective strategies for health promotion and increased health literacy have strong evidence-base. This would require multidisciplinary collaboration but a narrow range of PsyCap training may reduce stress to some degree but does not equip individuals with additional resources to limit their risk of ill-health, particularly chronic conditions or NCDs.

Minor

1. Fix typo in line 71, it should be major and not mayor.

2. Use the term “sex” rather than “gender”. I notice LISS uses the term gender but am unable to review its operational definition. It is therefore not clear whether LISS still uses a binary or has expanded the categories to include intersex/transex people. Perhaps a footnote explaining this would be helpful.

3. Please add some known limitations of online surveys. This could be an extract from LISS panel survey or your own observations.

6. PLOS authors have the option to publish the peer review history of their article (what does this mean?). If published, this will include your full peer review and any attached files.

Reviewer #1: No

Reviewer #2: No

Reviewer #3: No

Reviewer #4: No

Reviewer #5: Yes: Rafat Hussain

---

## [Author Response · Author response to Decision Letter 0]

14 Oct 2020

August 27th, 2020

Dear dr. John-Henderson, 

Please find enclosed, the revised version of our research paper to PLOS ONE as an original study, entitled: ‘Explaining socioeconomic inequalities in self-reported health outcomes: The mediating role of perceived life stress, financial self-reliance, psychological capital, and time perspective’. We would like to thank you and the reviewers for their thorough examination and their constructive, relevant remarks and suggestions. We feel that the quality of our paper has improved by the changes we have made according reviewers’ comments. We have now completed all the changes requested. To summarize the main requested changes made in the revised manuscript: 

• We included more information about our dependent variables and a more thorough discussion of Marmot’s work in the Introduction section.

• The Introduction was also revised in such a way the aim of the study and resulting and tested hypothesis have been made clearer.

• We included a paragraph about the context/country in which the study takes place earlier in the Introduction section and made the scope of our study more clear. 

• We performed new analyses in which separate indicators for SEP are included (highest achieved educational level, personal income and employment status), background/origin of participants is added as a control variable at every level of the tested model and a tangible reserve (financial self-reliance) is included as an additional mediator in the model. We also adjusted the Results and Discussion section accordingly. 

• We included a more thorough and broader discussion of implications in our Discussion section.

We hope that the changes we have made resolve your concerns about the publication of our manuscript. In the text below, we respond to all of the reviewers’ comments point by point, and indicate how we have changed the manuscript accordingly. As requested, we re-submitted a marked-up copy and an unmarked (clean) file of our revised manuscript. All authors approved the revised manuscript and have contributed significantly to the work. We are more than willing to make any further changes if necessary to facilitate successful publication.

Thank you once again for your time and interest in our work. We look forward to hearing from you.

Yours sincerely,

Dr. Karen Schelleman-Offermans

Department of Work and Social Psychology

Maastricht University

Karen.offermans@maastrichtuniversity.nl

Responses to reviewers

We would like to thank all the reviewers for the time to read our manuscript and their thoughtful feedback. In the next sections we respond to all the comments of reviewers point by point. Our reaction to reviewers are displayed in italic and the adjustments made in the revised manuscript are displayed between the following signs @@ .. @@. Revisions made in the Results section are, not displayed here, since we fully revised the text. For these adjustments, we would like to refer the reviewers and the editor to the revised document.

Reviewer #1

This study has some interest in analyzing the mediating role of psychological factors in the relationship between SocioEconomic position (SEP) and self-rated health.

However, the methods and the data do not support the conclusion: «Psychological capital (hope, efficacy, resilience, optimism) and perceived life stress showed to mediate the SEP-Health gradient and therefore play a large role in explaining it (543-545). There is a lack of coherence between the theory and the methods resulting in the omission of social variables, which paves the way for a psychological explanation of the relationship under study.

The premise of the author’s argumentation is that resources to change structural factors through policies (education, income, migration status, etc.) requests too many resources, so there is a need to concentrate on individuals and psychological factors. Is there a good reason to elude fundamental factors that are well known and demonstrated in the literature on social inequities in health (please see the commission on the social determinants of health): living and working conditions, social exclusion, migration status, racism and other discriminations, access to health care, etc.

The approach of the authors reduce social inequalities to psychological inequalities between individuals. In addition to this too individualistic approach, it has been shown that groups are affected by social inequalities. Furthermore the social gradient in health demonstrate that inequalities are relative to the position in the social structure, that related to power relationship. Please see social epidemiology (Lynch and Kaplan, Social position and health, in Berkman and Kawachi, 2000)

The concept of ‘reserve capacities’ are defined by authors as composed of three dimensions: « tangible reserves (e.g., money and transportation). intrapersonal reserves (e.g., self-esteem and psychological capital) and interpersonal reserves (e.g., social support).» (59-62). However, authors are declaring the following:

«To sum up then, the literature on the RCM indicates that psycho-cognitive variables, or reserve capacities, play a mediating role in explaining the (indirect) effect of SEP on various health outcomes, and as such illustrates the usefulness of the RCM in understanding the psychosocial pathways leading to socioeconomic health inequalities.(95-98).

The following independents variables are included: SEP, stress, time, psychological capital. But what about interpersonal and tangible reserve? What about migration status, region of birth, living and working conditions, discrimination, social network, and access to health care ?

In fact, the model that is tested is made of psychological model.

Authors state in the discussion that « the results were largely in line with our expectations, and provide further support for the RCM » (lines 444) and that the current model does not fully capture the underlying mechanisms explaining the SEP-health gradient (lines 448-49). Indeed, the RCM was only partly tested (intrapersonal reserves), and the important social factors that are cited above were not taken into account. In most WHO models individual factors are located at the end of the model, after macro and living and working conditions.

I would also disagree with the proposed intervention focusing on individuals. As if, finally, the causes of social inequalities are psychological and the intervention is also psychological.

I do not want to underestimate the psychological dimension of human beings, but the explanation of deleterious effects of socioeconomic inequalities cannot be reduced to individuals’ psychology.

First, we would like to thank the reviewer for his/her feedback and input to improve the manuscript. We agree with the reviewer that the way the manuscript was presented did not justify the importance of social variables in explaining health inequalities. It was not our aim to reduce perceptions of health inequalities to psychological inequalities and we certainly acknowledge the importance of fundamental social factors that have shown to influence social inequalities in health. Although the scope of the current study was to specifically look at the role that stress, psycho-social reserves and time perspective play in explaining perceptions of health inequalities, we agree that we should have also included more fundamental social factors. Therefore, we conducted new analyses in which we included background/origin of participants (Dutch, non-Dutch Western or non-Dutch non-Western) as a control variable at all levels of the model and we included three separate indicators for socio-economic position (highest achieved educational level, being in paid employment and personal disposable monthly income). Furthermore, financial self-reliance was added to the model as a tangible reserve and additional mediator. We changed the Introduction section, including more background information regarding more fundamental social factors (Marmot’s work and work of Lynch as suggested by the reviewer) and included more fundamental social factors, as much as possible, now in the new analysis. The Results and Discussion sections were changed accordingly.

When including the above described variables into the model, new results showed that highest achieved educational level and being in paid employment seem to play a larger role in the social stratification within self-reported and self-perceived health outcomes than personal disposable income. Also, the psychological factors perceived life stress and psychological capital are still highly significant mediators regarding the association between highest achieved educational level and self-reported health outcomes, which indicates that our previous conclusion still stands. Furthermore, the new results showed that also tangible reserves play a role in explaining health inequalities. Changes made in the revised manuscript are depicted below. The Results section was completely revised and we refer for those changes to the revised document. 

Changes made in the Introduction section 

(lines 47-64)

@@ There are social, political and economic forces that shape the nature of social and structural relations in the society in which we live resulting in an unequal distribution of money, power and resources [e.g., 1–3]. Education, income and wealth are indicators of what resources individuals hold and what sort of ‘life chances’ they have [e.g., 1–3]. These structural positions are powerful determinants of the likelihood of health-damaging exposures and of possessing particular health enhancing resources. For instance, a higher income allows increased access to better quality material resources (e.g., food, housing, access to services) and a higher educational level provides individuals with greater knowledge and skills which can provide a higher social standing, self-esteem and facilitate participation in society [1]. The most commonly used indicators for socio-economic position (SEP) are therefore a) income level, b) educational level, and c) paid employment. Indeed, there are large differences in health behaviours between individuals within advantaged groups with a higher SEP and people within less advantaged groups with a lower SEP, leading to high disparities in self-rated health, health status as well as life expectancy amongst these groups [4–7]. To decrease the social stratification of health inequities, the obvious fundamental option is to change its structural drivers such as decreasing inequities in power, money and resources determined by the macro socioeconomic and political context [2]. Nevertheless, another option is to change specific risk or protective factors mediating the effect of SEP on health. @@

(lines 150-154)

@@ A tangible reserve that may be important in explaining differences in subjective health perceptions between individuals scoring lower and higher on different SEP indicators is how financially self-reliant a person is. A higher financial self-reliance (e.g., being able to pay your bills independently) may indicate higher tangible reserves and in turn, in accordance with the RCM, may increase positive cognitions and emotions and increase health outcomes.

 @@

(lines 121-125)

@@ Inferring from the theoretical assumptions of the RCM (see Fig 1), the main aim of the current study was to investigate whether the negative association between indicators of a low SEP and subjective self-reported health outcomes can be explained by higher perceived life stress, lower intrapersonal and tangible reserves and consequently more negative or positive time perspectives (cognitions).@@

Changes made in the Method section 

(lines 302-314)

@@Three indicators for SEP were used in the current study; a) highest achieved educational level, b) monthly personal disposable income and c) paid employment (0/1). Highest achieved educational level was measured on a 6-point scale (1=primary school, 2=intermediate secondary school/junior high school, 3=higher secondary education/senior high school, 4=intermediate vocational education/junior college, 5=higher vocational education/college, 6=University). Mean scores were used in the analyses and higher mean scores indicate a higher achieved educational level.

Monthly personal disposable income was asked on a 12-point scale ranging from 0 (no income) to 12 (having an income of > 7500). The mean score of this scale was used in the analyses. A higher mean score indicates a higher monthly personal disposable income. 

Paid employment was derived from a question on what the main daily activity of participants was with being in paid employment of one of the answer categories. Being in paid employment was thereafter dichotomized (no/yes).@@

(lines 348-353)

@@Financial self-reliance

Financial self-reliance was measured using responses to the item “To what extent do you have trouble with your money matters, such as paying bills and keeping eye on expenditure?”; ranging from 1= “A lot of trouble, I can only realize it with help” to 5= “no trouble at all, I do not need help”. The mean score of the reversed recoding of this item was used in the analyses to indicate a person’s financial self-reliance. @@

(lines 386-389)

@@In all analyses, and at all levels in the model, sex (Men=1; Women=2), age and background (included as two dummy variables: a) Western non-Dutch vs. non-western and Dutch; b) non-Western non-Dutch vs. Western and Dutch) were included as covariates@@

Changes made in Discussion section

(lines 565-588)

@@ First, our results showed that highest achieved educational level and being in paid employment seem to play a larger role in the social stratification within self-reported and self-perceived health outcomes than personal disposable income. Controlling for all other model variables, personal disposable monthly income did not show an association with self-reports on physical health conditions and self-perceived health (only an association with negative time perspective orientations).

Secondly, although there are significant and expected mediating paths from SEP indicators to both self-reported health-related outcomes, this effect was only found for the SEP indicator highest achieved educational level. Being in paid employment did show a significant association with a higher self-perceived health when controlling for all other model variables, nevertheless, this association was not mediated by the included mediators in the model. Furthermore, a direct effect of highest achieved educational level (SEP indicator) on self-reported health conditions and self-perceived health remained significant after including the mediators in the analyses and controlling for several other important covariates. This is in line with previous research in European countries [e.g., 10,44], and it suggests that the current model does not fully capture the underlying mechanisms explaining the SEP-health gradient. Our results do indicate that perceived life stress, and reserves (PsyCap and financial self-reliance) play a significant (mediating) role in explaining why individuals with a lower achieved educational level (SEP indicator) show lower self-perceived health and higher self-reports of physical health conditions, compared with individuals with a higher achieved educational level. Our results thus suggest that the higher experienced stress and lower reserves (psychological capital – hope, optimism, resilience and efficacy – and financial self-reliance) could make individuals with a lower SEP more vulnerable for perceived health conditions or perceiving their health as poor.@@

(lines 650-666)

@@ Nevertheless, although social stratification is associated with differences in perceived stress and psycho-social reserves, we should not be blind to the fact that social stratification is also a characteristic of a society in which inequities in power, money and resources are the structural drivers of health inequities [2]. The socioeconomic and political context is important in explaining health inequities. We should therefore bear in mind that individual indicators of SEP are derived from social and economic processes that shape the distribution of education, occupations and income across the population. The fact that, in the Netherlands, teachers have lower expectations of children with a low SEP, resulting in lower support from primary and secondary schools for their attempts to access tertiary education even when they have satisfactory test scores [14], is a serious cause for concern. In order to decrease the SEP-Health gap, in line with our results, especially teachers and schools should be educated to decrease this stigma for children with a low SEP to narrow this growing opportunity gap in the Netherlands. Furthermore, results of our study indicate that to decrease self-reported physical health conditions, political action is needed to increase employment, especially under people with a low SEP. Future research should also focus on differences in regions or countries where variety in these macro factors can be investigated that may further elucidate the mechanisms behind the SEP-Health gradient.@@

(Lines 718-729)

@@ To conclude, this study successfully extents current scientific literature by providing more insight into the role psychological variables play in the SEP-Health gradient. Psychological capital (hope, efficacy, resilience, optimism), financial self-reliance and perceived life stress showed to mediate the association between a lower achieved educational level and self-reported health outcomes and therefore play a large role in explaining this association. Furthermore, this study provides practical knowledge to develop timely prevention efforts aimed at reducing the SEP-Health gradient. Prevention efforts trying to reduce the SEP-Health gradient should focus on simultaneously increasing PsyCap and financial self-reliance and increasing coping levels with perceived life stress for individuals with a low achieved educational level. Furthermore, prevention efforts should additionally focus on narrowing the opportunity gap in education between children with a low versus a higher SEP (e.g., by educating teachers and schools how to prevent stigma) and reduce unemployment among people with a low SEP.@@

 

Reviewer #2: Review: Explaining socioeconomic inequalities in self-reported health: The mediating role of perceived life stress, psychological capital, and time perspective. – PONE-D-20-03476.

Comments to the authors

General comment:

This study investigates what role perceived life stress, psychological capital and time perspectives play in explaining socioeconomic inequalities in self-reported perceived health and physical health complaints. Overall, the paper is well written, the data are valuable, the scope of the study is interesting and the SEM analytic strategy is appropriate. This being said, there are some minor issues that need to be addressed.

Specific comments:

1. Authors should avoid the use of terms like “rates” when describing self-reported health outcomes along the manuscript. Similarly, they should not refer to “effects” when testing associations in cross-sectional analysis.

2. Line 174-80. Hypotheses H6a, H6b, H7a and H7b are not shown in Figure 1, as the authors mentioned in the text. Please, correct this.

3. Line 182-183. Please, specify that you refer to a direct association between low SEP with more negative self-reported health outcomes.

4. I assume that for methodological reasons the authors dichotomized the SEP indicator. This involves a loss of information about the intermediate spectrum of the SES gradient. Authors should mention this as a limitation of the study. Further, they should state in the discussion the potential bias in self-reported health outcomes associated with gender and age.

5. Line 260-262. “The Short Form of the Zimbardo Time Perspective Inventory [31], derived from the original Zimbardo Time Perspective Inventory [18], was used to measure the five different time perspectives (present-fatalistic, present-hedonistic, past-negative, past-positive).” Please, note that four time perspectives are indicated.

6. Table 1. Please, do not include “mean” in the description of the variables of Table 1. Mean and SD are already indicated in the columns.

We thank the second reviewer for his/her review and complements on the manuscript as it already stands and the suggestions to even further improve the manuscript. Below we respond to the points raised separately. 

1. We agree with the reviewer that we should not refer to effects when testing cross-sectional associations. We changed this throughout the Results and Discussion section. 

2. We adapted Figure 1 as suggested by the reviewer and added hypotheses 6a, 6b, 7a and 7b.

 

Revised Figure 1:

3. As suggested by the reviewer, we changed Line 182-183 accordingly. 

Revised line 182-183 in previous Introduction section (now line 227-230)

@@First, we hypothesize that indicators of a lower SEP (i.e., a lower achieved educational level, lower personal disposable income and not being in paid employment) are directly associated with more negative self-reported health outcomes (higher self-reports of physical health conditions and lower perceptions of general health; H1).@@

4. We originally dichotomized the SEP indicator and combined two measures (educational level and income) into one SEP indicator in the previous version of our manuscript. We realize that this may have resulted in a loss of information, also on the intermediate spectrum of the SEP gradient. Therefore, in the revised manuscript we included three separate SEP indicators including as much variance within the different indicators as possible.

Furthermore, we added in the Discussion section the possible bias in self-reported health outcomes associated with age and gender as suggested by the reviewer. We believe that by adding age and gender as covariates at every level of the model, that this possible bias is reduced. Nevertheless, we also recommend future research to include a larger sample in which gender and age specific analyses are possible and more insight can be given into gender and age specific results. 

Changes made in the Method section 

(lines 302-314)

@@Three indicators for SEP were used in the current study; a) highest achieved educational level, b) monthly personal disposable income and c) paid employment (0/1). Highest achieved educational level was measured on a 6-point scale (1=primary school, 2=intermediate secondary school/junior high school, 3=higher secondary education/senior high school, 4=intermediate vocational education/junior college, 5=higher vocational education/college, 6=University). Mean scores were used in the analyses and higher mean scores indicate a higher achieved educational level.

Monthly personal disposable income was asked on a 12-point scale ranging from 0 (no income) to 12 (having an income of > 7500). The mean score of this scale was used in the analyses. A higher mean score indicates a higher monthly personal disposable income. 

Paid employment was derived from a question on what the main daily activity of participants was with being in paid employment of one of the answer categories. Being in paid employment was thereafter dichotomized (no/yes).@@

Revisions made in the Discussion section 

(lines 677-685) 

@@ Secondly, a broad age range (16-96 years) was used in the current study, nevertheless, SEP indicators and time perspectives may change or develop over time in a persons’ life and therefore may have a different effect in a different stage of their lives [57]. Also, no separate analyses for men and women were performed, nevertheless men and women may deal with stress in a different way or may think differently about the future. We did control for sex and age at every level of our model to minimize the effect this could have had on our results. Future research should use a bigger sample to be able to conduct age and gender specific group analyses to be able to define more specific implications for health promotion, prevention and management for different subgroups.@@

5. We indeed only mentioned 4 time perspectives instead of 5. We changed this section as suggested by the reviewer and added the fifth time perspective.

Revised line 260-262 in previous manuscript (now line 316-318)

@@The Short Form of the Zimbardo Time Perspective Inventory [39], derived from the original Zimbardo Time Perspective Inventory [24], was used to measure the five different time perspectives (future, present-fatalistic, present-hedonistic, past-negative, past-positive).@@

6. As suggested by the reviewer we excluded mean in the description of the variable names in this table (in the revised version of the manuscript this is Table 2). 

Revised Table 2:

@@ Table 2. Mean scores (SD, mean std. errors) of model variables for paid employment (SEP indicator)

 Paid employment

 No Yes

Self-perceived health 

Range 1-5 2.87*** (.82, .04)

N=355 3.20 (.70, .05)

N=172

Self-reported physical health conditions 

Range 0-8 2.18*** (1.99, .11)

N=355 1.35 (1.36, .10)

N=172

Perceived life stress 

Range 0-4 1.63 (.59, .03)

N=326 1.57 (.55, .05)

N=153

Psychological capital 

Range 1-6 4.20*** (.81, .05)

N=326 4.49 (.74, .06)

N=153

Financial self-reliance 

Range 1-5 3.77** (.62, ,03)

N=348 3.91 (.38, .03)

N=170)

Present-hedonistic time perspective

Range 1-5 3.06 (.86, .05)

N=326 3.12 (.84, .07)

N=152

Present-fatalistic time perspective 

Range= 1-5 3.11* (.84, .05)

N=326 2.94 (.85, .07)

N=152

Past-negative time perspective 

Range 1-5 2.40** (1.13, .07)

N=326 2.10 (.97, .08)

N=152

Past-positive time perspective

Range 1-5 3.89 (.81, 05)

N=326 3.93 (.83, .07)

N=152

Future time perspective 

Range 1-5 3.86 (.69, .04)

N=326 3.92 (.60, .05)

N=152

Personal monthly disposable income

Range 0-11 2.28** (1.79, .09)

N=400 4.26 (1.94, .14)

N=197

This is the Table 2 legend. SEP = socio-economic position; * p < .05; ** p < .01; *** p < .001.@@

Reviewer #3: This paper has the ambition to extend literature on (perceptions of) health by studying whether

psychological capital and feelings of stress mediate the impact of socio-economic inequality. This is an interesting and relevant topic to study but unfortunately, there are fundamental theoretical and conceptual flaws throughout the paper. The authors study a good number of relevant associations and competently test these, but need to provide the same level of clarity in embedding their paper in the relevant literature and in supporting their conceptual choices. Below, I outline my main comments:

We thank the third reviewer for his/her review and complements on the manuscript as it already stands and the suggestions to even further improve the manuscript. Below we respond to the points raised separately. 

1. In the introduction, when discussing the social gradient, the discussion on what socio-economic position exactly entails and how it is important is superficial. The authors do no define what socio-economic position is and overlook the decades of research on social stratification where a distinction is made between social class, status, income inequality, different levels of capital,…

The authors could start by looking at an older overview by Bergman & Joye (2001) and more

recent work by Weeden, Wodtke, Oesch,…

1. We agree with the reviewer that we should have included a more elaborately what SEP entails and how it can influence health in the introduction section. We therefore included more elaboration in the introduction section regarding this issue. We also thoroughly looked at the suggested literature. We decided to include other references (work of Marmot and Lynch) to elaborate on what SEP entails and how it can influence health. 

Revisions made in the Introduction section

(lines 47-60)

@@There are social, political and economic forces that shape the nature of social and structural relations in the society in which we live resulting in an unequal distribution of money, power and resources [e.g., 1–3]. Education, income and wealth are indicators of what resources individuals hold and what sort of ‘life chances’ they have [e.g., 1–3]. These structural positions are powerful determinants of the likelihood of health-damaging exposures and of possessing particular health enhancing resources. For instance, a higher income allows increased access to better quality material resources (e.g., food, housing, access to services) and a higher educational level provides individuals with greater knowledge and skills which can provide a higher social standing, self-esteem and facilitate participation in society [1]. The most commonly used indicators for socio-economic position (SEP) are therefore a) income level, b) educational level, and c) paid employment. Indeed, there are large differences in health behaviours between individuals within advantaged groups with a higher SEP and people within less advantaged groups with a lower SEP, leading to high disparities in self-rated health, health status as well as life expectancy amongst these groups [4–7]. To decrease the social stratification of health inequities, the obvious fundamental option is to change its structural drivers such as decreasing inequities in power, money and resources determined by the macro socioeconomic and political context [2]. Nevertheless, another option is to change specific risk or protective factors mediating the effect of SEP on health.@@

2. The problem statements is not fully clear until the end of the introduction. In the first few pages, the authors discuss life expectancy, health behavior, and health inequalities almost as synonyms. In health sociological and epidemiological research, however, different mechanisms explain disparities regarding these distinct concepts (see the work of Marmot, Singh-Manoux, Demakakos,…). Since the dependent variable ultimately seems to be subjective health perceptions, it is imperative that the authors both clarify this early on and in addition conceptually disentangle this from other concepts related to health.

2. We now, more clearly, state the main aim and outcome measures of the study, earlier in the introduction section and specified it later on, also referring to the included Fig 1. Also, we included how subjective or perceived health measures relate to other concepts related to health and that our study focuses on subjective or perceived health outcomes. 

Revised Introduction section 

(lines 70-79)

@@The main aim of the current study is to gain more insight into the psychological mechanisms explaining the SEP-health gradient. The current study utilizes the reserve capacity model (RCM) [14] as a theoretical basis to gain insight into the psychological mechanisms of health inequalities in a general Dutch sample. More specifically, this study investigates the role of perceived life stress, perceived intra-personal (psychological capital) and tangible (financial self-reliance) reserves and time perspective in explaining differences in self-reported physical health conditions and self-perceived health in individuals with a lower and higher SEP. Self-rated or perceived health covers a variety of health outcomes and has shown to be consistently associated with morbidity and mortality [5,6]. This suggests that a single measure of self-rated health is a strong predictor of overall health status.@@

(lines 121-125)

@@ Inferring from the theoretical assumptions of the RCM (see Fig 1), the main aim of the current study was to investigate whether the negative association between indicators of a low SEP and subjective self-reported health outcomes can be explained by higher perceived life stress, lower intrapersonal and tangible reserves and consequently more negative or positive time perspectives (cognitions).@@

3. Arguing that (perceived) health inequalities exist solely because of differences in resources is too simple and ignores a broad literature on the interplay between health-related choices, (lack of) resources, cognitive differences, stress,… in understanding health inequalities.

3. The reserve Capacity Model does not explain health inequalities solely through differences in resources, but incorporates the relations also with cognitive differences and stress in understanding health inequalities. We also included stress and cognitions into our model. We do realize that this may not have been completely made clear in our previous version. Therefore we adapted the text sections regarding this in the Introduction section. 

Revisions in Introduction section 

(lines 102-105)

@@ The reserve capacity model is a framework explaining how a lower SEP results in health disparities over time, and it specifically explicates the mediating role of ‘reserve capacities’, stress and positive/negative cognitions and emotions in explaining the SEP and health status gradient [9,15–17].@@

Furthermore, we do realize that in the current study, we focus more on psychological factors underlying health inequalities and that more fundamental social factors are important as well (for specific revisions see our reaction to reviewer 1). Therefore, we conducted new analyses in which we included background/origin of participants (Dutch, non-Dutch Western or non-Dutch non-Western) as a control variable at all levels of the model and we included three separate indicators for socio-economic position (highest achieved educational level, being in paid employment and personal disposable monthly income). Furthermore, financial self-reliance was added to the model as a tangible reserve and additional mediator. We changed the Introduction section, including more background information regarding more fundamental social factors (Marmot’s work, as also mentioned by other reviewers) and also included more fundamental social factors, as much as possible, now in the new analysis. We also changed the analysis and the Results and Discussion sections accordingly.

4. Especially problematic throughout the manuscript is the constant confusing of reality and

perceptions thereof. Since the authors study self-reported health and assume a mediation of psychological capital on the pathway from SEP to self-reported health it is problematic that they do not pay attention to the distinction between those concepts that are measured in an objective manner and those that tap into subjective perceptions (of health). An emerging sociological/epidemiological literature has illustrated that these need to be disentangled since different mechanisms cause material and subjective differences, while both have a disparate impact on health outcomes.

4. We understand that perceptions of health may not be the same as actual or more objectively measured health outcomes. In the revised version of our manuscript we now consistently talk about perceptions or self-reported health outcome measures, as also suggested by reviewer 2. We also included in our revised Discussion section that, although self-rated or perceived health covers a variety of health outcomes and has shown to be consistently associated with morbidity and mortality [5,6], the mechanisms explaining health inequalities regarding more objective measured health outcomes might be different.

Revisions made in the Discussion section 

(line 709-715)

@@ Moreover, this study explicitly looked at subjectively measured health outcome measures. Although self-rated or perceived health covers a variety of health outcomes and has shown to be consistently associated with morbidity and mortality [6,7], the mechanisms explaining health inequalities regarding more objective measured health outcomes might be different. Future research on elucidating the mechanisms behind health inequalities should therefore also include more objectively measured health outcomes.@@

5. From line 175, the authors start with introducing hypotheses 6a-7b, while the other hypotheses

are introduced later. While this is a strange order to say the least, the problem here mainly lies

in the fact that the hypotheses often are not logically deducted from the literature. This is due

to the fact that the literature review frequently confuses distinct concepts and/or ignores

important strands of the literature on social inequality and health (perceptions).

5. We understand that this may have been confusing and that it seemed like we start describing hypotheses 6a-7b from line 175 in the previous manuscript. Nevertheless, this was the footnote of Figure 1, which needs to be displayed after the title of the Figure in the manuscript. The description of the hypotheses starts at line 182 of the previous manuscript (in the revised version line 224) and starts with describing hypothesis 1. As already noted in our reaction to point 1, 2, and 4 of this reviewer, we also included more information about other stands of the literature on social inequality and health including the description of the influence of more structural social factors.

6. From line 254 the authors introduce the usage of income and educational background as

indicators of socio-economic position. No theoretical support of this choice is provided while the

introduction does not include any discussion of the literature using such a conceptualization of

socio-economic inequality. Why would income and/or educational background be a valid way of

measuring socio-economic inequality?

6. As also mentioned in previous reactions, in the revised version of our manuscript, we included a more detailed description on what socio-economic inequality entails which makes it also easier to understand why we use personal income, highest achieved educational level and employment status as indicators for SEP (for revisions in the manuscript see reactions to point 1 of reviewer 1 and point 1 of reviewer 3). As also mentioned in previous reactions, we now include three separate SEP indicators (for revisions see point 4 of reviewer 2) instead of using both income and educational level into one SEP dichotomous variable. 

 

Reviewer #4: Thank you for the opportunity of reviewing the manuscript ‘Explaining socioeconomic inequalities in self-reported health: The mediating role of perceived life stress, psychological capital, and time perspective’. The manuscript addresses an important topic but I believe it is currently not suitable for publication in PlosOne.

Overall, the strength of the paper lies in the very comprehensive description of the psychological constructs, the mediating model and the discussion of such mediating role. However, in order for the manuscript to make a contribution the authors should integrate what is a very psychologically oriented study within the broader understanding of health disparities derived from sociology, economics or social policy.

This is very apparent in how the authors consider socio-economic position and how they conceive the psychological constructs.

We thank the fourth reviewer for his/her review and complements on the manuscript as it already stands and the suggestions to even further improve the manuscript. Below we respond to the points raised separately. 

1. The authors consider differences in self-reported health by socio-economic position and the mediating role of three psychological constructs: stress, psychological capital and time discounting. The conceptualisation of socio-economic position (SEP) in the manuscript is poor. It remains unclear why the authors discuss SEP rather than socio-economic status (SES). Operationally, SEP which is a latent construct is considered coarsely through a high-low conceptualisation based on arbitrary thresholds of disposable net income and educational attainment. This is a crucial flaw that invalidates all the analyses. This methodological problem is compounded by the failure of the author/s to discuss how and why SEP may shape individual health not only indirectly through its role in determining stress, psychological capital and time perspectives but also directly. A really large literature examines socio-economic disparities in health and the authors do not discuss/report any of this. They simply acknowledge in the discussion section that direct ‘residual’ and statistically significant association ‘remains’.

1. We agree with the reviewer that the conceptualization of SEP in our previous version was not adequate. We changed the SEP indicator in the revised version of our manuscript by including three separate SEP indicators (highest achieved educational level, employment status and personal disposable monthly income). This resulted in new analyses and results. We also included more literature on the more fundamental socio-economic disparities in health and discuss these points in our Discussion section. For specific revisions, please see our reaction to point 1 of reviewer 1. 

2. The hypotheses and review of the literature are not integrated. The review of theories and prior studies should be woven in the formulation of testable hypotheses.

2. We adjusted the structure of the revised Introduction section in such a way, that the aim and Figure (Fig 1) were hypotheses are displayed (which are the suggested relations of the Reserve Capacity Model) are now mentioned earlier in the Introduction section (for specific revisions see our response to reviewer 3 point 2) . Also, the structure of the Introduction and literature review, is now more in line with the different hypotheses of our study. We did, however, still include a last paragraph were specific hypotheses are displayed, also referring to Fig 1. Since there are quite some hypotheses in this study, due to the many relations that are tested in the model, we believe it is clearer to state the specific hypothesis altogether at the end of the Introduction section. Since there are many revisions made in the structure of the Introduction section, we refer to the revised document for the specific text revisions. 

3. We learn only on page 10 that the study is based on data from the Netherlands. Given that the study is about the role socio-economic position plays in shaping health outcomes, it would be important to situate the research within the broader economic, social, health and welfare policies that operate in the Netherlands vs., for example, the United States, the United Kingdom or other countries. Both overall levels of health disparities and which factors shape variability across social groups can be expected to vary depending on, just to name an obvious feature, if health care is provided free of charge to all or depends on employer provided health insurance.

3. We agree with the reviewer that more contextual information on the Netherlands should be given earlier in our manuscript. We therefore included a small paragraph in the Introduction section about this and also use this contextual information to inform practical implications. 

Revision made in the Introduction section

(Lines 80-98)

@@ The Dutch context

In the Netherlands, as in many other European countries, health inequalities between people with a high and low SEP exist; people with a low SEP show for instance a higher prevalence of chronic diseases and self-assessed poor health [10]. Also, individuals with a low SEP live on average 6 years shorter and 15 years with a less well-experienced health, compared with individuals with a high SEP in the Netherlands [11]. When looking at the more fundamental drivers of health inequalities, although being one the richest countries in the world, the Netherlands faces an income inequality of .28% (Gini coefficient) and a poverty rate of 8.8% [12]. Nevertheless, for European standards, the Netherlands has a relatively flat income distribution; the differences in disposable income between people are small. Also, overall unemployment (3.3% in 2017) is low in the Netherlands [13]. Although high quality education and student loans are available to every person growing up in the Netherlands, children of lower educated parents are less likely to go to university [14]. This, for some part, seems to be due to the lower expectations teachers have of these children and the resulting lower support schools give them in their attempts to access tertiary education, even when they have satisfactory test scores [14]. Furthermore, the Netherlands has a universal healthcare system, managed by the Dutch government and supplemented by private insurers. Everyone living or working in the Netherlands must obtain basic level health insurance from a Dutch provider. People with a low income have the right to apply for health care contribution supplied by the Dutch government. @@

Revision made in the Discussion section

(Lines 650-664)

@@ Nevertheless, although social stratification is associated with differences in perceived stress and psycho-social reserves, we should not be blind to the fact that social stratification is also a characteristic of a society in which inequities in power, money and resources are the structural drivers of health inequities [2]. The socioeconomic and political context is important in explaining health inequities. We should therefore bear in mind that individual indicators of SEP are derived from social and economic processes that shape the distribution of education, occupations and income across the population. The fact that, in the Netherlands, teachers have lower expectations of children with a low SEP, resulting in lower support from primary and secondary schools for their attempts to access tertiary education even when they have satisfactory test scores [14], is a serious cause for concern. In order to decrease the SEP-Health gap, in line with our results, especially teachers and schools should be educated to decrease this stigma for children with a low SEP to narrow this growing opportunity gap in the Netherlands. Furthermore, results of our study indicate that to decrease self-reported physical health conditions, political action is needed to increase employment, especially under people with a low SEP.@@

4. I suggest the authors discuss the characteristics of individuals who were excluded from the sample because of missing information. Why not use multiple imputation to recover them? This would also allow to have a larger sample size.

4. We are very grateful the reviewer made this suggestion. Although we already used Full Information Maximum Likelihood parameter estimates to test our model to impute the 4% missing data from the questionnaire filled out by participants in Feb-March, we actually did not think of imputing all the missing, using the data of the participants that did fill out the earlier questionnaire conducted by the Liss panel but did not fill out the questionnaire in Feb-March. In the revised version of the manuscript, we recovered and added the data of the people that were first excluded from the analyses and imputed the missings by using Full Information Maximum Likelihood parameter estimates. This enhances our analytic sample to 600. The characteristics of this sample is now described in the revised manuscript as well. 

Revision in the Method section

(lines 284-299)

@@The two datasets were combined into one cross-sectional dataset with a total sample of 600 participants whenever missings are imputed. 

Participants

In the analytic sample, a total of 255 men (42.5%) were present in the sample. The mean age was 53.4 years (SD=18.79; range 16-96) for the complete analytic sample. No significant difference (F=6.64, p=.10) in mean educational level was found between men and women. There were significantly more men in paid employment than women (χ2=10.96, df=1, p=.001). . Also, participants in paid employment significantly (F=63.2, p<.001) differed in mean age compared with participants not in paid employment; 45.9 (SD=11.8, mean std. error=.8) for participants in paid employment and 57.1 (SD=20.4, mean std. error=1.0) for participants not in paid employment. The mean educational level did not significantly differ (F=1.04, p=.35) for people from different backgrounds (Dutch, Western, non-Dutch and non-Western). Also, background did not significantly differ (χ2=3.6, df=2, p=.2) for participants in paid employment (77.4% Dutch, 12.3% non-Dutch Western, 10.3% non-Dutch non-Western) or not in paid employment (82.5% Dutch, 7.5% non-Dutch Western, 10.0% non-Dutch non-Western).@@

 

Reviewer #5: This is an important topic which needs ongoing research given the increasing socioeconomic and health disparities.

There are several areas where the current paper would benefit from changes and improvements.

We thank the fifth reviewer for his/her review and complements on the manuscript as it already stands and the suggestions to even further improve the manuscript. Below we respond to the points raised separately. Since point 1 and 2 refer to similar issues, as well as point 5-6 we combined our reaction to these point of this reviewer. 

1. Although the authors cite the work of Michael Marmot as the first two references, it is unclear whether they have fully understood his central arguments or is it the way the study questions are drafted that leaves room for concern. Marmot and many others have repeatedly pointed out that health conditions experienced by people, particularly chronic health conditions are predicated on several risk factors that occur over the lifespan and are often beyond the control of the individuals. In fact, many of the SES [or SEP] inequities stem from systemic macro-level discriminatory policies and programs that often favor those already advantaged over the poor. It would be helpful to state these more clearly and expand on these important set of issues.

2. There is often a tendency for researchers to look at cause-effect relationship using psycho-social variables to assess individual level variability but with little acknowledgement of the larger issues. The current paper falls into the same trap where there is an inherent assumption that “individuals have/can have control over their lives” and “are in a position to improve their health by positive thinking” – be it in the form of Reserve Capacity Model/theory or Psychological Capital. The downside of such assumptions is that it leads to unbalanced discussion of underlying factors that produce health and ill-health. It would be useful for the authors to review the paper and redraft some sections with more open and upfront 

acknowledgment of the limitations and problematic implications of the theories they have used.

1-2. We agree with the reviewer that the underlying factors produce health and ill-health should have also been discussed in the manuscript. We added more information on these underlying factors in the Introduction and Discussion section. Because this issue was also mentioned by other reviewers, we would like to refer to our reaction to point 1 of reviewer 1 for specific text revisions that have been made in the revised manuscript. 

3. It would be useful to bring in some definitional issues about SEP and cut-offs. The authors have used a particular definition (education and income) but without explaining why two metrics were chosen. For an international audience, some context would be helpful.

3. We understand that the cut-offs might have seemed arbitrary. Therefore, as also mentioned to other reviewers raising the same issue, we now included new analyses in which SEP is included as three separate indicators (personal disposable monthly income, highest achieved educational level and employment status) not using any predisposed cut-offs. Since the Results section has been completely revised due to the changes in the analysis, we refer to the revised manuscript for specific revisions.

4. I looked up the LISS data archive but am unable to correlate the time-frame used by the study with the Waves listed in the LISS Data Archive. It would be useful to include some information on this aspect.

4. We agree that this was not clear in our previous version and now included the wave to which the data collection period responds to.

Revisions made in the Method section

(Lines 267-269)

@@ Self-perceived health and self-reported physical health conditions were collected during the regular panel measurements in November-December 2018 (Health wave 11).@@

5. Although the study has controlled for age in the final analyses, but the biological issues of a huge age range of 16-92 remains problematic. Very few young people (<30) in lower SEP would have health conditions that lead to poor health rating and their outlook would be generally positive. On the other hand, many older people would experience a number of chronic health conditions over the lifespan and for a 70 or 80-year old these issues get compounded and confounded. A more nuanced description in the Introduction with relevant references and a more balanced set of statements in the Discussion section would be of value.

6. Perhaps it would be useful to use some right censoring to limit very old age groups to limit the confounding caused by increased frailty and multimorbidity. Whilst this may sound “ageist”, it makes little sense from a population health perspective to include such a large age range as the implications for action are very different in terms of health promotion, prevention and management. I realize this is not the central focus of the paper but this the reality of individuals in the study sample; and has significant implications for population level health issues.

5-6 We agree with the reviewer that using a broad age range could be problematic, nevertheless, we did control for age on every level in the model, which should have reduced confounding. Age specific group analyses would have reduced the samples in such a way that this was not possible in the current study with all the variables and relationships included in the model. Nevertheless, we did add this concern in our limitation section (for text revisions in our Discussion section see point 4 of reviewer 2) and also suggest future research should use bigger samples were age-specific group analyses are possible and can inform implications for health promotion, prevention and management in a more specific way.

7. Sampling. The authors outline stratified random sampling based on age, gender [sex] with oversampling of LISS sample for low SEP. It is not clear how this then remains a random sample and what adjustment were made in the analyses to adjust for oversampling. These issues are important if population level inferences are to be drawn from the study.

7. We agree with the reviewer that this is a point of concern and therefore Discuss this in the Limitation section of our revised manuscript.

Revision in the Discussion section

(lines 671-676)

@@ First, it is important to state that individuals scoring lower on SEP indicators were oversampled in our study to ensure sufficient variability in the data collected. Thus, although our results give useful insights into the relative health disadvantage of people scoring lower on the measured SEP indicators, our results do not elucidate the public health importance of the socioeconomic health inequality in terms of the size of the exposed population or absolute level of risk. @@

8. I find use of the term “health complaint” highly problematic. Perhaps it is disciplinary differences but when respondents have been asked about health issues many of which are almost diagnostic categories [except for headache and fatigue], then a more appropriate term would be health conditions. These are not psychosomatic problems but actual health conditions. Reviewing LISS variables they are listed as health problems but include a combination of the both categories. A few short explanatory sentences should suffice.

We agree with the reviewer that the term health complaint might problematic and changed this term to health conditions throughout the complete manuscript. 

9. With regards to Implications, I was somewhat disappointed that there is no mention of improving health literacy. Whilst macro-level change to reduce inequities [SEP] are harder for researchers to influence, effective strategies for health promotion and increased health literacy have strong evidence-base. This would require multidisciplinary collaboration but a narrow range of PsyCap training may reduce stress to some degree but does not equip individuals with additional resources to limit their risk of ill-health, particularly chronic conditions or NCDs.

We now included health literacy into our implication section as an important other factor that could influence health outcomes. 

Revisions made in Discussion section 

(lines 704-709)

@@ Moreover, the effect of low achieved educational level on the self-reported health outcomes used in the current study was only partly mediated by lower levels of PsyCap, financial self-reliance and higher perceptions of life stress. Therefore, more research is needed to gain insight into other factors that can explain why low SEP indicators result in worse self-perceived health outcomes. For instance, previous research has indicated health literacy as part of the mechanism explaining the relationship between education and health [58]. @@

Minor

1. Fix typo in line 71, it should be major and not mayor.

We adjusted this typo. 

2. Use the term “sex” rather than “gender”. I notice LISS uses the term gender but am unable to review its operational definition. It is therefore not clear whether LISS still uses a binary or has expanded the categories to include intersex/transex people. Perhaps a footnote explaining this would be helpful.

We changed the term gender into sex throughout the complete document. LISS still uses a binary variable for sex, we describe this in the Method section (line 387: sex (Men=1; Women=2)).

3. Please add some known limitations of online surveys. This could be an extract from LISS panel survey or your own observations.

Known limitations of online surveys are low response rates, non-randomized samples and non-participation of people who do not have access to internet. Nevertheless, participants of the LISS panel are randomly sampled from the population sample of the Netherlands. Furthermore, sampled people who do not have access to the internet or do not have a device to actually fill out the questionnaire, are given free access to internet and a free device. Therefore, response rates using the LISS panel are high (almost 80%). This information is already in the Method section lines 256-260. 

@@The panel is based on a true probability sample of households drawn from the population register. Households that could not otherwise participate are provided with a computer and Internet connection [38].@@

---

## [Decision Letter · Decision Letter 1]

12 Nov 2020

PONE-D-20-03476R1

Explaining socioeconomic inequalities in self-reported health outcomes: The mediating role of perceived life stress, financial self-reliance, psychological capital, and time perspective.

PLOS ONE

Dear Dr. Schelleman-Offermans,

Thank you for submitting your revised manuscript to PLOS ONE. After careful consideration, we feel that it has merit but does not fully meet PLOS ONE’s publication criteria as it currently stands. Therefore, we invite you to submit a revised version of the manuscript that addresses the points raised during the review process.

As you will see, the reviewer feels that the revised manuscript is greatly improved. However, they identify a few necessary minor revisions require your attention. 

We look forward to receiving your revised manuscript.

Kind regards,

Neha John-Henderson

Academic Editor

PLOS ONE

Reviewers' comments:

Reviewer's Responses to Questions

**Comments to the Author**

1. If the authors have adequately addressed your comments raised in a previous round of review and you feel that this manuscript is now acceptable for publication, you may indicate that here to bypass the “Comments to the Author” section, enter your conflict of interest statement in the “Confidential to Editor” section, and submit your "Accept" recommendation.

Reviewer #5: All comments have been addressed

2. Is the manuscript technically sound, and do the data support the conclusions?

Reviewer #5: Yes

3. Has the statistical analysis been performed appropriately and rigorously? 

Reviewer #5: Yes

4. Have the authors made all data underlying the findings in their manuscript fully available?

Reviewer #5: Yes

5. Is the manuscript presented in an intelligible fashion and written in standard English?

Reviewer #5: Yes

6. Review Comments to the Author

Reviewer #5: The revised manuscript reads well for most parts. A few minor revisions need to be undertaken.

Abstract. The Result section would benefit from review and rephrasing of key findings. In its current form, it can be misunderstood.

Introduction. Lines 145-150

1. Need to briefly explain the mediating pathway for lower BMI and Cholestrol levels.

2. Reference 22 is not the most appropriate paper to cite here. Reading through the paper by Rew et al (2016), its Pyscap intervention focus in on young homelessness women with substance abuse issues. These are not the main features of your study participants. Therefore need to replace this reference with another more relevant one, as the statement in your manuscript provides potentially misleading information. Alternatively revise the statement.

Typos

Lines: 225-227. Review font.

Line 448: Review for typo.

References

Reference # 35 is incomplete. Need to provide relevant information.

7. PLOS authors have the option to publish the peer review history of their article (what does this mean?). If published, this will include your full peer review and any attached files.

Reviewer #5: No

---

## [Author Response · Author response to Decision Letter 1]

12 Nov 2020

Responses to reviewers

Reviewer #5: The revised manuscript reads well for most parts. A few minor revisions need to be undertaken.

We thank reviewer 5 for his time to carefully read the manuscript and the constructive feedback to further improve our manuscript. In the sections below we address the minor issues raised by reviewer 5 point by point. 

1. Abstract. The Result section would benefit from review and rephrasing of key findings. In its current form, it can be misunderstood.

We agree with the reviewer that the Result section of the Abstract was unclear. We revised this accordingly.

Revised section of the Abstract: 

@@ Results: Highest achieved educational level and being in paid employment showed to play a role in the social stratification within self-reported and self-perceived health outcomes, whereas this was not found for personal monthly disposable income. The association between a lower highest achieved educational level and lower self-perceived health was mediated by lower PsyCap and higher perceived life stress levels. The association between a lower highest achieved educational level and higher levels of self-reported physical health conditions was mediated by less financial self-reliance and higher perceived life stress levels. Although no mediating role was found for time perspective orientations in the association between the measured SEP indicators and health outcomes,negative time perspective orientations were associated with either self-perceived health or self-reported physical health conditions. 

2. Introduction. Lines 145-150

a. Need to briefly explain the mediating pathway for lower BMI and Cholestrol levels.

We realized that the illustration of the results of the study by Luthans et al. was not clearly explained. In fact, this study tested no mediation pathways including BMI and cholesterol levels directly. Neverthless, this study included analyses in two separate steps, where first the association between health-related PsyCap with BMI, cholesterol and health satisfaction was tested. In a separate analyses, this study found that health satisfaction was associated with overall well-being. In the revised version of the introduction we now more clearly explain these results. 

Revised Introduction section (Lines 148-150).

@@For example, Luthans et al. [21] showed that increased health-related PsyCap is associated with lower BMI and cholesterol levels, as well as with health satisfaction. In turn, in a separate analysis, health satification showed a significant association with overall well-being in this study.@@

b. Reference 22 is not the most appropriate paper to cite here. Reading through the paper by Rew et al (2016), its Pyscap intervention focus in on young homelessness women with substance abuse issues. These are not the main features of your study participants. Therefore need to replace this reference with another more relevant one, as the statement in your manuscript provides potentially misleading information. Alternatively revise the statement.

The reference was mentioned just to illustrate that PsyCap can be developed and changing in PsyCap may also lead to changes in health-related behavioral outcomes. We added more information about the study in the Introduction section to avoid to be potentially misleading the reader. 

Revised Introduction section (Lines 150-158).

@@ Also, PsyCap has been shown to be open for development. For instance, in a study by Rew, Powell, Brown, Becker & Slesnick [22], the feasibility and efficacy of a brief psychological capital intervention was examined using a quasi-experimental pre-post research design with repeated measures. The brief intervention aimed to reduce health-risk behaviors (alcohol use and sexual risk behavior) in 80 ethnically diverse homeless women by increasing their psychological capital. Study results showed that within this group, substance use decreased significantly over time whereas safe-sex self-efficacy and behaviors significantly increased over time [ 22].@@

3. Typos

Lines: 225-227. Review font.

Line 448: Review for typo.

The font in lines 225-227 was reviewed and adjusted accordingly as well as the typo in Line 448.

Revised line 448 (in the revised version this is line 481)

In line with hypothesis 3a, the reserves PsyCap and financial self-reliance were positively associated with better self-reported health outcomes (higher self-perceived health and lower self-reported physical health conditions; hypothesis 3a, see Table 1).

4. References

Reference # 35 is incomplete. Need to provide relevant information.

We provided the complete information for this specific reference in the revised version of our manuscript.

Revised reference

@@35. Bosma H, Schrijvers C, Mackenbach JP. Socioeconomic inequalities in mortality and importance of perceived control: cohort study. BMJ. 1999;319: 1469–1470. Available: www.bmj.com@@

---

## [Editor Report · Decision Letter 2]

26 Nov 2020

Explaining socioeconomic inequalities in self-reported health outcomes: The mediating role of perceived life stress, financial self-reliance, psychological capital, and time perspective.

PONE-D-20-03476R2

Dear Dr. Schelleman-Offermans,

We’re pleased to inform you that your manuscript has been judged scientifically suitable for publication and will be formally accepted for publication once it meets all outstanding technical requirements.

Kind regards,

Neha John-Henderson

Academic Editor

PLOS ONE
---

## [Editor Report · Acceptance letter]

14 Dec 2020

PONE-D-20-03476R2 

Explaining socioeconomic inequalities in self-reported health outcomes: The mediating role of perceived life stress, financial self-reliance, psychological capital, and time perspective. 

Dear Dr. Schelleman-Offermans:

I'm pleased to inform you that your manuscript has been deemed suitable for publication in PLOS ONE. Congratulations! Your manuscript is now with our production department. 

Kind regards, 

on behalf of

Dr. Neha John-Henderson 

Academic Editor

PLOS ONE